# Vlasov simulation of electrons in the context of hybrid global models: An eVlasiator approach

Markus Battarbee[1], Thiago Brito[1], Markku Alho[1], Yann Pfau-Kempf[1], Maxime Grandin[1], Urs Ganse[1], Konstantinos Papadakis[1], Andreas Johlander[1], Lucile Turc[1], Maxime Dubart[1], and Minna Palmroth[1,2]

[1]Space Physics Research group, Department of Physics, University of Helsinki, Helsinki, Finland
[2]Finnish Meteorological Institute, Helsinki, Finland
**Correspondence:** Markus Battarbee (markus.battarbee@helsinki.fi)

**Abstract.** Modern investigations of dynamical space plasma systems such as magnetically complicated topologies within the Earth's magnetosphere make great use of supercomputer models as well as spacecraft observations. Space plasma simulations can be used to investigate energy transfer, acceleration, and plasma flows on both global and local scales. Simulation of global magnetospheric dynamics requires spatial and temporal scales achievable currently through magnetohydrodynamics or hybrid-
kinetic simulations, which approximate electron dynamics as a charge-neutralizing fluid.

We introduce a novel method for Vlasov-simulating electrons in the context of a hybrid-kinetic framework in order to examine the energization processes of magnetospheric electrons. Our extension of the Vlasiator hybrid-Vlasov code utilizes the global simulation dynamics of the hybrid method whilst modelling snapshots of electron dynamics on global spatial scales and temporal scales suitable for electron physics. Our eVlasiator model is shown to be stable both for single-cell and small-
scale domains, and the solver successfully models Langmuir waves and Bernstein modes. We simulate a small test-case section of the near-Earth magnetotail plasma sheet region, reproducing a number of electron distribution function features found in spacecraft measurements.

## 1   Introduction

Physical processes in near-Earth space are dominated by plasma effects such as non-thermal particle distributions, instabilities,
plasma waves, shocks, and reconnection. Modern research into space phenomena utilizes both spacecraft measurements and supercomputer simulations, investigating how ions, electrons, and electric and magnetic fields interact in the vicinity of plasma structures. Spacecraft provide point-like observations, limited in their ability to investigate spatial structures, although modern constellation missions can have multiple satellites close by allowing for multipoint analysis to decipher, e.g., current sheet directions (Escoubet et al., 2001; Burch et al., 2016a). Computer simulations on the other hand are limited by spatial resolution,
time stepping, and the large difference between ion and electron temporal and spatial scales (see, e.g., Tóth et al., 2017).

Simulations capable of modelling the whole near-Earth geospace have historically used magnetohydrodynamics, neglecting kinetic effects and implementing electrons only as e.g. the Hall term correction to Ohm's Law. These models can be run for extended periods of time, but as they model plasma motion as a fluid, they use coarse grids, e.g. $0.25\,R_{\rm E}$ (Janhunen et al.,

2012) or $0.1\,R_{\mathrm{E}}$ (Wang et al., 2020) (where $R_{\mathrm{E}}$ is the Earth radius) and cannot model kinetic effects but are sufficient for some global dynamics. Recent advances have allowed global investigations into hybrid-kinetic models, where ions are treated as a kinetic self-consistent species and electrons are a charge-neutralizing fluid. Successful approaches include hybrid-Vlasov models (Palmroth et al., 2018) and hybrid-PIC (particle-in-cell) codes (e.g. Lin and Wang, 2005; Sibeck et al., 2008; Omidi et al., 2009; Karimabadi et al., 2014). Kinetic investigation run times rarely exceed one hour or hundreds to a few thousand ion gyroperiods. The simulation spatial resolution is chosen to be relevant to the scales of investigation, with the most usual metric being the ion inertial length $d_i$. The simulation domain must encompass the necessary global dynamics with sufficient space to manage boundary effects.

In order to understand electron physics, kinetic modelling of electrons has been investigated by a number of methods such as full-PIC (ions and electrons as interacting particles, e.g., Hesse et al. 2005), full-Vlasov (ions and electrons as interacting distribution functions, e.g., Umeda et al. 2009; Schmitz and Grauer 2006; Pezzi et al. 2019), hybrid-PIC-electrons (dynamic electron particles, ions as a static background, e.g., Lapenta et al. 2007) and hybrid-Vlasov-electrons (dynamic electron distribution function with ions as a static background, e.g., Nunn 2005). In fully kinetic numerical investigations, the standard approach is to alter the ion-to-electron mass ratio of $\sim 1836$ to, e.g., 50 (Hesse et al., 2005) or 25 (Wilson et al., 2016) in order to achieve interesting dynamics with available computational resources. Using explicit solvers, resolving waves and kinetic electron instabilities to prevent simulation self-heating requires the spatial resolution to encompass the Debye length $\lambda_D$ (Birdsall and Langdon, 2005) and the time stepping must resolve the electron plasma oscillation $\omega_{\mathrm{pe}}$. This can, however, be bypassed via semi-implicit or implicit solver methods. If such an approach is used and the resolution is decreased, selecting a very low resolution may result in the loss of some electron physics. Effects such as the Dungey cycle (Dungey, 1961), involving the whole magnetosphere, are unachievable with traditional kinetic electron approaches. Full-PIC approaches have, however, been applied to investigations of, e.g., reconnection in a Harris current sheet (Harris 1962, investigated in, for example, Lapenta et al. 2015; Daughton et al. 2011) or asymmetric reconnection (Hesse et al., 2016). Pritchett (2000) presents a historical review of magnetospheric PIC simulations and anticipates the development of more realistic, global 3-D magnetosphere models with increasing computational resources.

More recent simulation studies of electron physics in the magnetosphere such as the PIC simulations by Bessho et al. (2014, 2016) and Hesse et al. (2016) have focused on local regions, modelling for example electron diffusion regions (EDRs) and extracting resultant electron velocity distribution functions (eVDFs). Liu et al. (2013) investigated the small-scale three-dimensional structure of EDRs with a realistic proton-electron mass ratio with a small configuration, and extended to a larger local 3-D configuration with a reduced proton-electron mass ratio. These simulations are always local with prescribed driving. A more global approach, MHD-EPIC, is presented by Daldorff et al. (2014), with a two-way coupling of a global, 2-D Hall MHD magnetosphere model and local implicit PIC model at regions of interest, where a proton-electron mass ratio of 25 was used. Notably, these PIC regions handled by implicit solvers do not resolve the Debye length. MHD-EPIC has since been employed to study the magnetosphere of Ganymede in 3-D with large embedded PIC domains by Tóth et al. (2016); Zhou et al. (2019).

An example of small-scale global, electromagnetic implicit PIC modelling for a weak comet has been performed by Deca et al. (2017, 2019) with a reduced proton-electron mass ratio of 100, and local simulations for a lunar minimagnetosphere (Deca et al., 2015) with a reduced proton-electron mass ratio of 256.

Ricci et al. (2002) discuss the effect of the ratio between the proton mass $m_\mathrm{p}$ and the electron mass $m_\mathrm{e}$ as a part of the GEM challenge, concluding that reconnection rates are well captured by smaller mass ratios of $m_\mathrm{p}/m_\mathrm{e} = 180$, although with modified electron kinetics. Lapenta et al. (2010) discusses modifications to electron microphysics at reconnection sites in more detail in relation to proton-electron mass ratios of 64, 256, and 1836 using an implicit PIC model.

Another approach compared to PIC simulations is to represent particle velocity distributions with moments beyond the MHD approach (Wang et al., 2015). For example, Huang et al. (2019) have developed a six-moment multi-fluid full-Maxwell model. They note that they do not capture reconnection to an acceptable accuracy and have yet to publish global simulation results. Global ten-moment results for the Hermean magnetosphere have been published by Dong et al. (2019). Furthermore, approaches which focus on electron effects at lower frequencies (neglecting effects at plasma oscillation timescales) have been investigated by, for example, Lin and Chen (2001) and Tronci and Camporeale (2015).

Several processes that occur in the magnetosphere that depend on electron behavior are still poorly understood. Recently, missions such as Magnetospheric MultiScale (MMS; Burch et al., 2016a) have enabled plasma measurements that are able to better resolve electron-scale physical processes. MMS in particular has provided data to many publications on magnetic reconnection (e.g., Burch and Phan, 2016; Phan et al., 2018; Huang et al., 2018; Hoilijoki et al., 2019a; Fargette et al., 2020), the most popular topic of electron physics investigations. Reconnection-driven jets and dipolarization fronts cause magnetic flux pileup and excitation of waves such as whistlers, affecting energy conversion and dissipation (Khotyaintsev et al., 2011; Breuillard et al., 2016; Zhang et al., 2018). Bulk flows along the tail lead to electrons heating up as they approach the Earth (Runov et al., 2015; Artemyev et al., 2013) with the electron-to-proton temperature ratio approaching even 1 (Wang et al., 2012). These flows interact with strong currents found in the plasma sheet (Nakamura et al., 2008; Artemyev et al., 2017). The dynamics of electrons near the current sheet include strong Hall fields and current sheet thinning (Lu et al., 2019, 2016). Electron anisotropies can excite electron-driven waves and time-domain structures, such as have been observed recently in different regions of the magnetosphere (e.g., Cattell et al., 2005; Mozer et al., 2015; Ergun et al., 2016). They have been characterized as whistler mode waves, electrostatic solitary waves and lower hybrid waves among other types. These waves interact strongly with electrons, causing effects such as heating, changes to temperature anisotropy, and particle energization. These energized electrons can then add to energetic particle precipitation, leading to the generation of auroras (Ni et al., 2016).

This paper introduces an alternative, novel method for simulating electron distribution function physics in the context of global ion-determined fields. The aim is to investigate how much of the global electron physics and distribution functions can be understood by utilising ion-generated field as modelled by hybrid-kinetic codes, as opposed to a numerically unfeasible global full-kinetic approach. The paper is organised as follows. In Section 2, we introduce the ion-kinetic hybrid-Vlasov code Vlasiator and how the Vlasov equation is solved. In section 3 we introduce the eVlasiator modifications implemented for the analysis of electron distribution functions. Section 3.1 describes how our electron simulation is set up from fields and moments modelled by an ion-kinetic simulation. Section 3.2 describes the time propagation of the eVDF and Section 3.3 details the

field solver changes implemented. Section 3.4 describes the sample test simulation used in this study. In Section 4 we perform rigorous validation and stability tests for our electron solver, and in Section 5 we present some electron distribution functions achieved by running our solver on a test dataset, comparing them with existing literature. Finally, Section 6 draws conclusions of our analysis and lays out a plan for future research approaches.

## 2 The Vlasiator ion-kinetic hybrid-Vlasov code

Vlasiator (von Alfthan et al., 2014; Palmroth et al., 2018) is, at the present time, the only hybrid-Vlasov code capable of simulating the global magnetospheric system of the Earth, accounting for ion-kinetic effects on spatial and temporal scales which model both magnetopause and magnetotail dynamics. Vlasiator solves the Vlasov equation for particle distribution functions discretized on Cartesian grids, with closure provided by Ohm's Law augmented by the Hall term. Each particle population is described using a uniform Cartesian three-dimensional velocity space grid (3V) with a resolution chosen to accurately model the solar wind inflow Maxwellian distribution and with extents chosen to encompass heated ion populations associated with the magnetosheath and flux transfer events. A standard Vlasiator global run proton velocity-space grid has a resolution of $30\,\mathrm{km\,s^{-1}}$, extending between $\pm 4020\,\mathrm{km\,s^{-1}}$. To constrain computational cost and memory usage, those parts of the velocity distribution function which have a phase-space density below a sparsity threshold are discarded, except for buffer regions which allow the correct growth of the VDF in these parts (von Alfthan et al., 2014). The proton sparsity threshold is usually set to a value between $10^{-17}$ and $10^{-15}\,\mathrm{m^{-6}\,s^3}$.

In the spatial domain, Vlasiator can be run in 1D, 2D, or 3D, with 2D the most usual choice in order to evaluate global dynamics. Simulations have used spatial resolutions of, e.g., $\Delta x = 228\,\mathrm{km}$ or $\Delta x = 300\,\mathrm{km}$, enough to accurately model ion cyclotron waves though not resolving the ion inertial length in all regions of the simulation domain. Large-scale global 3D runs will be made possible in the near future by adaptive mesh refinement (AMR), using non-uniform cell sizes in the spatial domain, thus cutting down on the computational cost of the simulation.

Vlasiator models standard collisionless space plasmas dominated by protons but can also model other particle species in the same self-consistent simulation. However, until now, the electron population has been treated in the usual way of implementing it as a massless charge-neutralizing fluid. The method does not track the evolution of electrons beyond assuming charge neutrality, and therefore, these standard Vlasiator simulations cannot be used to infer electron dynamics. This paper presents a novel approach for investigating how a global plasma current structure can influence electrons with limited self-consistency enforced through plasma oscillation and current densities.

## 2.1 Solving the Vlasov equation

Vlasiator uses the hybrid-Vlasov ion approach to model the near-Earth space plasma environment. The full six-dimensional (6D) phase space density $f_s(\boldsymbol{x}, \boldsymbol{v}, t)$, with $\boldsymbol{x}$ the ordinary space variable, $\boldsymbol{v}$ the velocity space variable, and $t$ the time variable, for each ion species $s$ of charge $q_s$ and mass $m_s$ is evolved in time using the Vlasov equation (1). The electric and magnetic fields, denoted $\boldsymbol{E}$ and $\boldsymbol{B}$ respectively, are evolved using Maxwell's equations: Faraday's Law (2), Gauss's Law (3) and Am-

père's Law (4), in which $\mu_0$ and $\varepsilon_0$ are the vacuum permeability and permittivity, respectively, and $\boldsymbol{j}$ is the total current density. The solenoid condition in Gauss's Law (3) is ensured via divergence-free magnetic field initial conditions reconstruction (Balsara, 2009). In the hybrid approach, electrons are assumed to maintain plasma neutrality, resulting in the charge density $\rho_q$ in Gauss's Law vanishing. In the Darwin approximation, also used in many hybrid codes, propagation of light waves is neglected by removing the displacement current term $\varepsilon_0 \frac{\partial \boldsymbol{E}}{\partial t}$ in Ampère's Law (4). The Vlasiator field solver follows the staggered-grid approach of Londrillo and Del Zanna (2004), and is described in detail in Palmroth et al. (2018).

$$\frac{\partial f_s}{\partial t} + \boldsymbol{v} \cdot \frac{\partial f_s}{\partial \boldsymbol{x}} + \frac{q_s}{m_s}\left(\boldsymbol{E} + \boldsymbol{v} \times \boldsymbol{B}\right) \cdot \frac{\partial f_s}{\partial \boldsymbol{v}} = 0. \tag{1}$$

$$\nabla \times \boldsymbol{E} = -\frac{\partial \boldsymbol{B}}{\partial t} \tag{2}$$

$$\nabla \cdot \boldsymbol{B} = 0 \text{ and } \nabla \cdot \boldsymbol{E} = \frac{\rho_q}{\varepsilon_0} \tag{3}$$

$$\nabla \times \boldsymbol{B} = \mu_0 \left(\boldsymbol{J} + \varepsilon_0 \frac{\partial \boldsymbol{E}}{\partial t}\right) \tag{4}$$

The generalized Ohm's Law providing closure for the Vlasov system is

$$\boldsymbol{E} + \boldsymbol{V} \times \boldsymbol{B} = \frac{\boldsymbol{J}}{\sigma} + \frac{\boldsymbol{J} \times \boldsymbol{B}}{n_e e} - \frac{\nabla \cdot \mathcal{P}_e}{n_e e} + \frac{m_e}{n_e e^2}\frac{\partial \boldsymbol{J}}{\partial t}, \tag{5}$$

where $\boldsymbol{V}$ is the plasma bulk velocity, $\sigma$ is the conductivity, $e$ is the elementary charge, $n_e$ is the electron number density, and $\mathcal{P}_e$ is the electron pressure tensor. In hybrid approaches of collisionless plasma, we can assume high conductivity, neglecting the first term on the right-hand side. In the limit of slow temporal variations, the electron inertia term (the last term on the right-hand side) also vanishes. The remaining two terms on the right-hand side of the equation are the Hall term, $\boldsymbol{J} \times \boldsymbol{B}/(n_e e)$, and the electron pressure gradient term, $\nabla \cdot \mathcal{P}_e/(n_e e)$. In hybrid models, a true description of electron pressure is unavailable so it must be described via some approximation such as adiabatic, isothermal or polytropic electrons or a fixed ion-to-electron temperature ratio, or by neglecting the small electron pressure gradient term altogether. The standard ion-hybrid Vlasiator code supports isothermal fluid electrons but existing simulations have always set this temperature to zero. This along with assuming charge-neutrality (proton number density $n_p = n_e$) results in the ion-hybrid Vlasiator using the simplified MHD version of Ohm's Law with the Hall term included:

$$\boldsymbol{E} + \boldsymbol{V} \times \boldsymbol{B} = \frac{1}{e n_p \mu_0}(\nabla \times \boldsymbol{B}) \times \boldsymbol{B}. \tag{6}$$

As Vlasov methods do not propagate particles but rather evolve distribution functions, we now briefly explain the semi-Lagrangian method employed by Vlasiator (for a full description, see chapter 5.3.1 in Palmroth et al. 2018). Vlasiator propagates distribution functions of particles following the SLICE-3D method (Zerroukat and Allen, 2012) and utilizing Strang splitting with advection (also referred to as translation, the second term of Vlasov's equation 1) and acceleration (the third term of Vlasov's equation 1) calculated one after the other with a Leapfrog offset of $\frac{1}{2}\Delta t$. In this manuscript, $\Delta$ denotes steps on the full simulation grid and associated time step and $\delta$ is used to indicate calculations performed as sub-stepping. For each time step, a Vlasov acceleration is evaluated with time step length $\Delta t$ which is, amongst other things, limited to a maximal

Larmor orbit gyromotion rotation value, which is usually set to 22°. This value is constrained by the SLICE-3D shear approach, with values much above 22°resulting in unphysical deformation of the VDF and smaller values increasing computational cost of the simulation. For each acceleration step of length $\Delta t$, a transformation matrix is initialized as an identity matrix. The transformation matrix is first composed to apply the uniform electric field acceleration and the gyromotion due to the magnetic Lorentz force. Then, the transformation matrix is decomposed into three shear transformations. For a detailed explanation of

the approach see chapter 3.5.1 of Palmroth et al. (2018). The transformation matrix is incrementally built with substepping of $\delta t$ where each $\delta t$ corresponds to a 0.1° Larmor gyration, with the gyration step derived from convergence tests. Instead of applying linear acceleration by the motional electric field, a method similar to the Boris-push method (Boris, 1970) is applied, where first a transformation is performed to move to a frame in which the electric field vanishes, then the rotation is applied, and then a frame transformation back to the original frame is added. In the standard hybrid formalism, the frame without

an electric field is found via the MHD Ohm's Law with the Hall term included (6). This Hall frame estimates the frame of reference of electrons, assuming electrons generate a current density which corresponds to the local magnetic field structure, in accordance with Ampère's Law. After substepping is evaluated, the transformation matrix is applied to the gridded velocity distribution function by the SLICE-3D algorithm.

## 3    The eVlasiator global electron solver

In this section we introduce a novel method of simulating electron dynamics within the Earth's magnetic domain by building on the strengths of Vlasiator simulations. The method, called eVlasiator, focuses on the evolution of accurately modelled velocity distribution functions based on global plasma dynamics and structures evolved by the hybrid model. The spatial scales used in Vlasiator are not sufficient to resolve in detail small-scale phenomena such as electron-dominated reconnection, but this balances out with a realistic representation of global structures and asymmetries of the whole magnetosphere. The eVlasiator

model solves the Vlasov equation for electron distribution functions using mostly the same methodology as Vlasiator itself, but applies a simplified field solver, neglecting magnetic field evolution.

### 3.1    Simulation initialization

    Modelling the evolution of electron distribution functions in response to global magnetic field structures requires input from the large-scale fields and moments of a Vlasiator simulation of near-Earth space. In the eVlasiator approach, we read magnetic

field vectors and proton plasma moments for the chosen simulation domain and apply user-defined temperature scaling to generate initial Maxwellian electron velocity distribution functions. We do not model electrons throughout the whole global domain, choosing instead a region of interest to reduce the computational cost, though our method is designed to work with any subset of and up to the whole global domain. For the selected domain, we read in the Vlasiator ion-hybrid simulation proton moments, cell-face-average magnetic field components and cell-edge-average electric field components (the latter being used

by the staggered-grid field solving algorithm from Londrillo and Del Zanna 2004). Both protons and electrons for the eVlasiator simulation are initialized from the read moments as Maxwellian distribution functions, with electron bulk velocity selected so

that Ampère's Law (4) is fulfilled. Re-mapping input run Vlasiator proton VDFs as Maxwellians does not affect the simulation results as eVlasiator only considers the proton number density and bulk velocity for current density calculations and does not propagate the proton distribution functions, instead keeping their characteristics completely constant for the duration of the eVlasiator simulation. For each simulation cell, we use the Balsara (2009) approach for calculating cell-average volumetric magnetic fields and respective derivatives. The eVlasiator solver uses volumetric field derivatives for calculating $\nabla \times \boldsymbol{B}$.

## 3.2 The eVlasiator electron solver

eVlasiator solves the evolution of electron eVDFs similar to how Vlasiator simulates proton VDFs (for a detailed explanation, see Palmroth et al. 2018, in particular chapter 5.3.1). Solving the Vlasov equation (1) is split into two sections, translation and acceleration, with each of these steps performed in a staggered Leapfrog approach. This approach is described in Figure 1 with the first row indicating the spatial advection of electrons (translation) and the second row describing the effect of the Lorentz force on electrons through electric field acceleration and gyromotion. At time $t$ (or $t_0$ at the initial state) we have the 5D (2D-3V) or 6D (3D-3V) electron velocity distributions (and, by extension, moments) in the whole simulation domain as well as proton moments and volumetric magnetic fields. Proton and magnetic field data is as read from the Vlasiator simulation, and kept constant throughout the eVlasiator simulation. At simulation start, the leapfrog stepping is initialized with a half-length acceleration step (shown in red as step 0. in Figure 1).

During each translation step, as depicted in Figure 1 and described by the equation

$$\frac{\partial f_s}{\partial t}\bigg|_{\text{trans}} + \boldsymbol{v} \cdot \frac{\partial f_s}{\partial \boldsymbol{x}} = 0, \tag{7}$$

we perform a semi-Lagrangian spatial advection operation using second-order polynomial remapping, in an identical fashion as in regular Vlasiator. This is evaluated separately for each cell in the gridded electron distribution functions using the velocity for that cell, and is evaluated for one Cartesian direction at a time.

During each acceleration step, as depicted in Figure 1 and described by the equation

$$\frac{\partial f_s}{\partial t}\bigg|_{\text{acc}} + \frac{q_s}{m_s} \left(\boldsymbol{E} + \boldsymbol{v} \times \boldsymbol{B}\right) \cdot \frac{\partial f_s}{\partial \boldsymbol{v}} = 0, \tag{8}$$

we perform a semi-Lagrangian velocity space SLICE-3D update of the whole local distribution function, separately for each spatial cell. This method evaluates the acceleration due to electric fields (uniform movement in velocity space) and the rotation due to the Lorentz force magnetic component (a rotation in velocity space). The uniform movement and the rotation are composed into a transformation matrix. To apply the transformation with the SLICE-3D scheme, the matrix is then decomposed into three shear motions, one along each Cartesian velocity coordinate axis, and performed using semi-Lagrangian fourth order polynomial remapping, similar to how the regular Vlasiator Vlasov solver works. This approach is applicable as long as velocities are non-relativistic. For a detailed description, see chapter 5.3.1 of Palmroth et al. (2018).

Due to the inherent connection between rapid electron motion and the local electric field response, we update electric fields in tandem with electron acceleration. The approach is detailed in the next subsection.

## 3.3 The eVlasiator field solver

In the eVlasiator field solver we maintain static magnetic fields as read from the input Vlasiator simulation, only calculating electric field evolution. We model the electric field by including additional terms in Ohm's Law (5), allowing for the interaction of electron distribution functions with electron-oscillation electric fields. Whistler mode propagation is not included in this study. We do not include any electric field due to Gauss' Law. We will consider each term of the eVlasiator field solver separately:

- As we keep magnetic fields static, we do not implement Faraday's Law (2).

- Collisionless plasma physics assumes that electrons are fast enough to balance out any charge imbalance, and in hybrid-kinetic simulations this holds true. We do not implement Gauss' Law (3) in order to quantify to what extent charge neutrality holds in the eVlasiator context.

- The last term in Ampère's Law (4) is the displacement current, which is neglected in the Darwin approximation. However, electron motion can be very rapid and thus we now include this term in our model, though still maintaining static magnetic fields. This approach thus constrains electrons to the defined static magnetic fields and does not introduce light waves.

- As our plasma remains collisionless, we maintain our assumption of infinite conductivity, and thus the $\boldsymbol{J}/\sigma$ term in the generalized Ohm's Law (5) remains zero.

- The Hall term, $\boldsymbol{J} \times \boldsymbol{B}/(n_e e)$, is used to estimate the electron reference frame. This term is no longer required, as the Lorentz gyromotion rotation can be performed in the actual electron bulk motion reference frame.

- As eVlasiator models electrons with full distribution functions, we include the full electron pressure tensor $\mathcal{P}_e$ and implement the electron pressure gradient term using spatial gradients calculated for electron pressure.

- The final term of the general Ohm's Law is the electron inertia term. Much like with our choice of including the displacement current, we now include the electron inertia term in our solver.

For electron dynamics to be modelled, electron gyration and plasma oscillation must both be considered. We choose to limit the acceleration time step $\Delta t$ to a maximum of $22°$ of Larmor rotation or $22/360$ of a single plasma oscillation. The value of $22°$ is used to ensure our VDF remapping algorithm SLICE-3D remains stable and the value $22/360$ was chosen for equal resolution of both oscillations as a result of convergence tests. Much larger values will result in neighboring simulation cells with different plasma characteristics diverging into an unstable state, and much lower values will needlessly cause an increase in computational cost. Due to the computational cost of SLICE-3D remapping, a substepping approach is used in order to more accurately model the electron gyromotion and plasma oscillation. Whilst the $22°$ step models eVDF evolution to a high accuracy, the accurate and stable simulation of feedback between electron velocity, plasma oscillation, and the electric field due to the electron inertia term in Ohm's Law requires substepping and places strict requirements on the length of the substep

$\delta t$. This substepping is performed in tandem with the SLICE-3D transformation matrix generation. The electron gyroperiod is
$\tau_{\mathrm{ce}} = 2\pi\omega_{\mathrm{ce}}^{-1}$ and the plasma oscillation time is $\tau_{\mathrm{pe}} = 2\pi\omega_{\mathrm{pe}}^{-1}$, where the electron plasma frequency is

$$\omega_{\mathrm{pe}} = \sqrt{\frac{n_{\mathrm{e}}e^2}{\varepsilon_0 m_{\mathrm{e}}}} \tag{9}$$

and the electron gyrofrequency is

$$\omega_{\mathrm{ce}} = \frac{eB}{m_{\mathrm{e}}}. \tag{10}$$

In transformation matrix generation, substepping is constrained to a maximum of $\delta t \leq \min(\tau_{\mathrm{pe}}, \tau_{\mathrm{ce}})/3600$. This value was
defined as a result of convergence tests, and its dependence on the relationship between $\tau_{\mathrm{pe}}$ and $\tau_{\mathrm{ce}}$ is discussed more in
section 4.

The electron oscillation and electric field feedback loop is handled in parallel with gyration by tracking a cell-volume-
averaged electric field $\boldsymbol{E}_{J_{\mathrm{e}}}$ which is itself derived from the small-scale electron oscillation. For each acceleration substep, we
update electron motion $\boldsymbol{V}$ and the electric field $\boldsymbol{E}_{J_{\mathrm{e}}}$ by performing two parallel 4th order Runge-Kutta propagations. The RK4
algorithm was chosen instead of a Runge-Kutta-Nyström method as it provides a good balance between general applicability,
stability, and computational performance. The first one is

$$\delta\boldsymbol{V}_{\mathrm{e}} = \delta t \frac{e}{m_{\mathrm{e}}} \boldsymbol{E}_{J_{\mathrm{e}}}, \tag{11}$$

tracking electron bulk velocity response $\delta\boldsymbol{V}_{\mathrm{e}}$ to the $\boldsymbol{E}_{J_{\mathrm{e}}}$ field. This simple acceleration term is in fact equal to evaluating
current changes via the electron inertia term in Ohm's Law with the $\boldsymbol{E}_{J_{\mathrm{e}}}$ field included in the left-hand-side electric field.
The second Runge-Kutta propagation tracks the evolution of the $\boldsymbol{E}_{J_{\mathrm{e}}}$ field due to changing current density, according to the
displacement current on the right-hand side of Ampère's Law (4) with the $\nabla \times \boldsymbol{B}$ term in Ampère's Law fixed to the static
input magetic fields. Thus, for each Runge-Kutta step, the electric field $\boldsymbol{E}_{J_{\mathrm{e}}}$ is updated with

$$\delta\boldsymbol{E}_{J_{\mathrm{e}}} = \delta t \left( \frac{\nabla \times \boldsymbol{B}}{\varepsilon_0\mu_0} - \frac{\boldsymbol{J}}{\varepsilon_0} \right) \tag{12}$$

$$= \delta t \left( \frac{\nabla \times \boldsymbol{B}}{\varepsilon_0\mu_0} - \frac{e\boldsymbol{V}_{\mathrm{p}}n_{\mathrm{p}} - e\boldsymbol{V}_{\mathrm{e}}n_{\mathrm{e}}}{\varepsilon_0} \right) \tag{13}$$

$$= \delta t\, c^2 \left( \nabla \times \boldsymbol{B} + \mu_0 e \left( n_{\mathrm{e}}\boldsymbol{V}_{\mathrm{e}} - n_{\mathrm{p}}\boldsymbol{V}_{\mathrm{p}} \right) \right) \tag{14}$$

where $c$ is the speed of light, and $\boldsymbol{B}$, $n_{\mathrm{p}}$ and the proton bulk velocity $\boldsymbol{V}_{\mathrm{p}}$ are assumed constant throughout the substep. Each of
the four $\delta\boldsymbol{V}_{\mathrm{e}}$ Runge-Kutta coefficients are updated with the latest estimate for $\delta\boldsymbol{E}_{J_{\mathrm{e}}}$, and vice versa. Values for $\boldsymbol{E}_{J_{\mathrm{e}}}$ are stored
between acceleration steps to ensure continuity of the oscillation. The change $\delta\boldsymbol{V}_{\mathrm{e}}$ calculated via each Runge-Kutta step is
then applied to the transformation matrix, allowing the solver to proceed to perform gyration in the electron frame of reference.
The substepping procedure is visualized in the third row of Figure 1. Further details of the solver and advection methods in
Vlasiator can be found in Palmroth et al. (2018).

With each substep, the transformation matrix is evolved by compounding the following transformations:

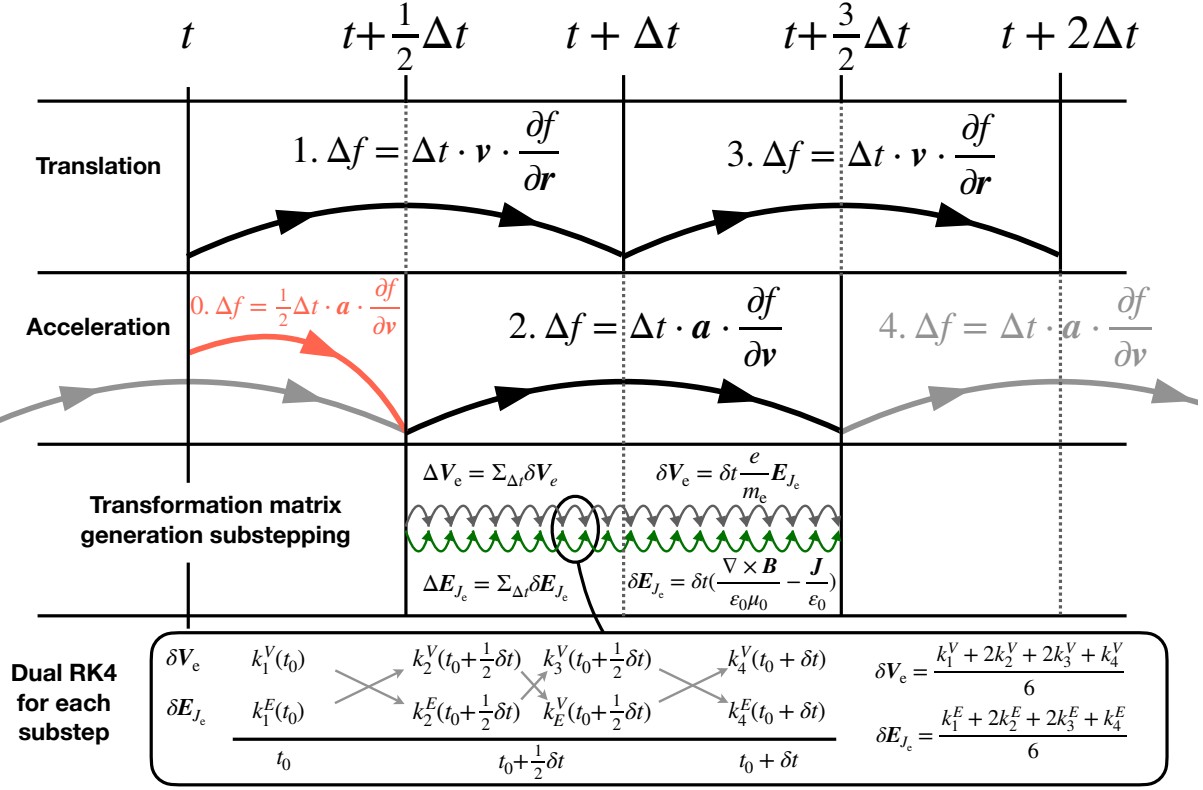

**Figure 1.** Electron solver procedure including substepping. At simulation start, a half-length acceleration step (0.) is performed. After that, translation (1,3,...) and acceleration (2,4,...) steps alternate in a Leapfrog approach. Each acceleration step applies a transformation matrix which is generated in substeps, each of which updates electron acceleration $\delta V_e$ and electric field change $\delta E_{J_e}$. Each of these updates is performed via a dual Runge-Kutta 4 algorithm over step lengths $\delta t$ with Runge-Kutta coefficients $k_{1...4}^E$ and $k_{1...4}^V$

1. Apply the acceleration $\delta V_e$ derived from RK4-substepped $E_{J_e}$ acceleration

2. Accelerate electrons by $\frac{1}{2}\delta t E_{\nabla \mathcal{P}_e}$

3. Transform to the frame of reference of the electron bulk motion

4. Rotate the eVDF around the magnetic field direction for $\delta t \omega_{ce}$

5. Transform back from the frame of reference of the electron bulk motion to the simulation frame

6. Accelerate electrons by $\frac{1}{2}\delta t E_{\nabla \mathcal{P}_e}$

After substepping is completed, the transformation matrix describing Vlasov acceleration is passed to the SLICE-3D algorithm, which decomposes the transformation into three Cartesian shears and updates the eVDF.

## 3.4 Sample simulation setup

In this method introduction, we use a noon-midnight meridional-plane 2D-3V Vlasiator simulation as our test-case input data. This 2D-3V Vlasiator simulation has been used to investigate global and kinetic magnetospheric dynamics in multiple studies such as Palmroth et al. (2017); Hoilijoki et al. (2017); Juusola et al. (2018a, b); Hoilijoki et al. (2019b); Grandin et al. (2019); Akhavan-Tafti et al. (2020). It has solar wind values of $\beta = 0.7$, magnetosonic Mach number $\mathrm{M_{ms}} = 5.6$, Alfvén Mach number $\mathrm{M_A} = 6.9$, proton number density $n_\mathrm{p} = 1\,\mathrm{cm^{-3}}$, and solar wind speed $\boldsymbol{u}_\mathrm{sw}$ along the $\hat{e}_x$ (Earth–Sun) direction with $u_\mathrm{sw,x} = -750\,\mathrm{km\,s^{-1}}$, simulating fast solar wind conditions and ensuring efficient simulation initialization. The simulation input interplanetary magnetic field is purely southward with $B_z = -5\,\mathrm{nT}$ and the Earth's magnetic dipole is a $\hat{e}_z$-aligned line dipole scaled to result in a realistic magnetopause standoff distance (Daldorff et al., 2014). The simulation has an inner boundary at $3 \cdot 10^6\,\mathrm{m} \approx 4.7$ Earth radii, modelled as a perfectly conducting sphere. The spatial resolution is $\Delta x = 300\,\mathrm{km}$.

For this eVlasiator sample run, we choose a region from the magnetotail with $70 \times 1 \times 40$ simulation cells in the X, Y, and Z directions, respectively. The subregion extent is from $x_- = -75.6 \cdot 10^6\,\mathrm{m}$ to $x_+ = -54.6 \cdot 10^6\,\mathrm{m}$, from $y_- = -0.15 \cdot 10^6\,\mathrm{m}$ to $y_+ = +0.15 \cdot 10^6\,\mathrm{m}$, and from $z_- = -6 \cdot 10^6\,\mathrm{m}$ to $z_+ = +6 \cdot 10^6\,\mathrm{m}$. Within this domain, visualized with a small rectangle in Figure 2a, the electron plasma period $\tau_\mathrm{pe}$ ranges from $\sim 0.7\,\mathrm{ms}$ in the magnetotail plasma sheet up to $\sim 2.5\,\mathrm{ms}$ in the near-plasmasphere lobes. The electron gyroperiod $\tau_\mathrm{ce}$ ranges from $\sim 14\,\mathrm{ms}$ in most of the lobes up to $\sim 770\,\mathrm{ms}$ at a tail current sheet X-line site.

The electron distributions are discretized onto eVlasiator velocity meshes, with the electron velocity mesh consisting of $400^3$ cells, extending from $-4.2 \cdot 10^7$ to $+4.2 \cdot 10^7\,\mathrm{m\,s^{-1}}$ in each direction, resulting in an electron velocity space resolution of $210\,\mathrm{km\,s^{-1}}$. The eVDF sparsity threshold was set to $10^{-21}\,\mathrm{m^{-6}\,s^3}$, ensuring good representation of the main structure of the eVDF. Discretizing a hot and dense electron distribution onto a Cartesian grid is numerically challenging without using vast amounts of memory. As portions of our simulation domain have proton temperature up to $10^8\,\mathrm{K}$, we use an empirical estimate of $T_\mathrm{i}/T_\mathrm{e} \sim 4$ as magnetosheath temperature ratios are usually around 4 to 12 (Wang et al., 2012). Paterson and Frank (1994), Hoshino et al. (2001), Artemyev et al. (2011), and Grigorenko et al. (2016) show similar proton-electron temperature ratios in the magnetotail. In order to constrain the extent of our velocity space and numerical requirements of our solver, we implement our electrons with a mass of 10 times the true electron mass, resulting in an ion-to-electron mass ratio of $m_\mathrm{i}/m_\mathrm{e} = 183.6$. As mentioned above, we calculate the required electron bulk velocity for each cell using the local volumetric (cell-average) derivatives so that the ion and electron fluxes in each cell correspond with the current density $\mathbf{J}$ required for fulfilling Ampère's Law (4) (with the displacement current neglected at initialization). This is equal to performing a transformation to the Hall frame of reference. Proton densities, magnetic field lines, proton temperatures, proton bulk velocities and electron bulk velocities calculated for simulation initialization are shown in Figure 2 along with an overview of the input Vlasiator simulation and the selected electron sub-domain.

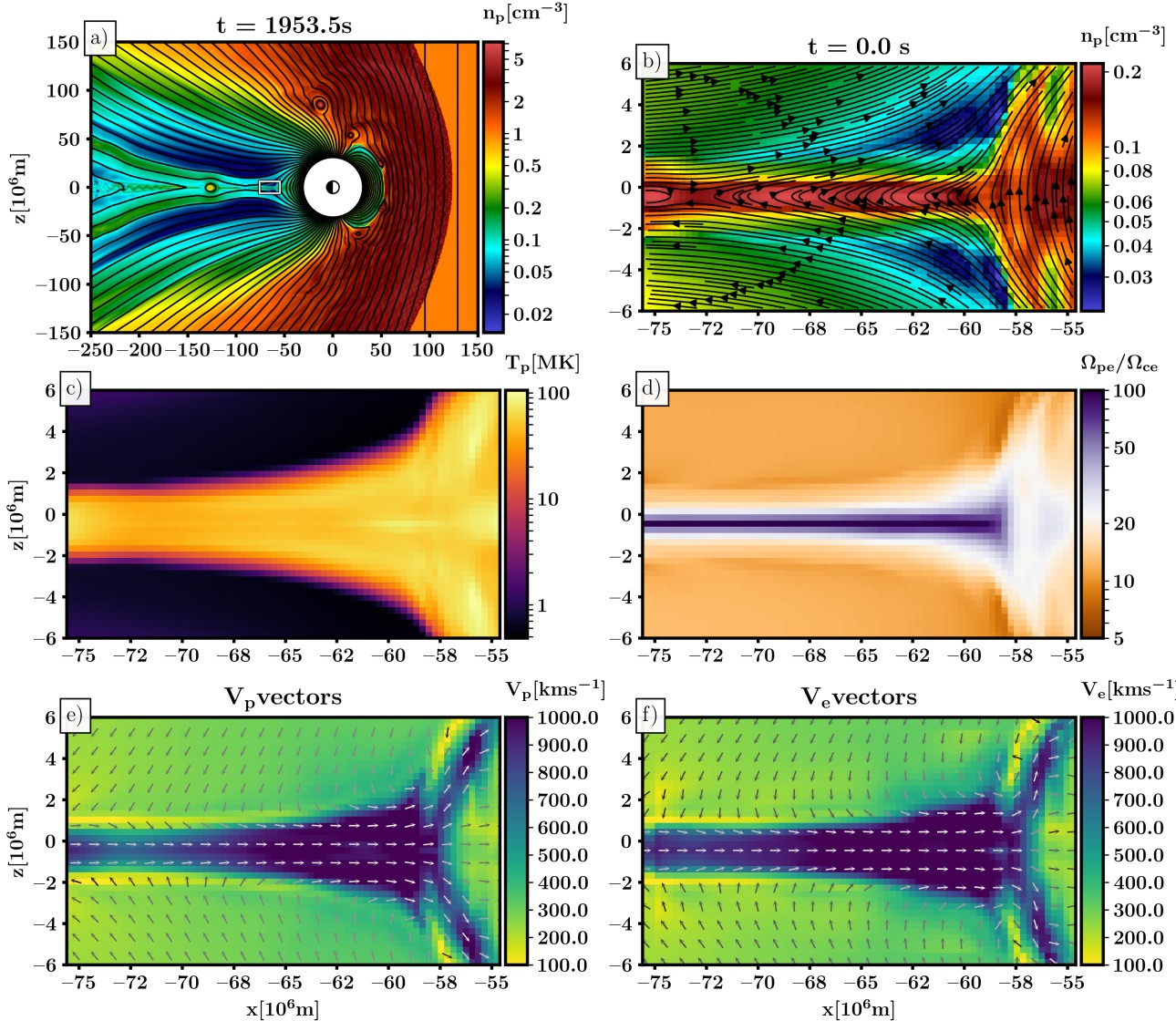

**Figure 2.** Simulation box initialization values. Panel a): Zoom-in to the central 16% section of the Vlasiator input simulation with plasma number density overlaid with magnetic field lines. A small rectangle in the magnetotail region indicates the electron simulation domain (panels b–f). Panel b): proton number density overlaid with magnetic field lines. X-line topology is visible at $X \sim -73 \cdot 10^6$ m $Z \sim -0.5 \cdot 10^6$ m. Panel c): Proton temperature as a scalar. Electron initialization temperatures are scaled down by a constant factor 4. Panel d): ratio of electron plasma and gyrofrequencies. Panels e) and f): Proton and electron bulk velocity magnitudes with in-plane directions indicated with vectors.

## 4 Solver performance

### 4.1 Single-cell stability of electron oscillation

To validate the performance of our electron solver, we performed single-cell tests, with resultant electron bulk velocities $\boldsymbol{V}_{\text{e}}$ and plasma oscillation electric fields $\boldsymbol{E}_{J_{\text{e}}}$ shown in Figure 3. These single-cell tests did not have magnetic field curvature or an ion population present, resulting in the electron motion oscillating around a stability point at $\boldsymbol{V}_{\text{e}} = 0$ and $\boldsymbol{E}_{J_{\text{e}}} = 0$. We set the electron number density to $n_{\text{e}} = 0.1\,\text{cm}^{-3}$ and the magnetic field to $B_x = 20\,\text{nT}$ (panels a through d) or $B_x = 200\,\text{nT}$ (panels e and f). We set an initial velocity perturbation of $\boldsymbol{V}_{\text{e},0} = (-100, -150, 200)\,\text{km}\,\text{s}^{-1}$, close to but below our electron velocity resolution of $\Delta v = 210\,\text{km}\,\text{s}^{-1}$. As can be seen from Figure 3, the electron oscillatory motion is well resolved and remains stable over an extended period. In panels e) and f) where the magnetic field strength was artificially increased in order to set the plasma and gyroperiods to values closer to each other (1.11 ms and 1.79 ms, respectively), we see a gradual evolution of oscillation amplitude and, thus, $\boldsymbol{E}_{J_{\text{e}}}$ field magnitude as the two types of electron motion interact. Over longer periods of time this growth becomes unstable, but it can be counteracted by using a smaller substep. Also, this instability occurs only when $\tau_{\text{ce}} \approx \tau_{\text{pe}}$ which does not occur in our full simulation domain.

### 4.2 Dispersion relation analysis

Although our method is geared towards solving electron motion at coarse spatial resolutions, to further validate the solver, a wave dispersion test was run (Kilian et al., 2017; Kempf et al., 2013). As waves are a collective, emergent phenomenon of the kinetic simulation approach, a correct reproduction of wave dispersion behaviour is a good indicator of correct physical behaviour of the simulation system.

Two 1D-simulation setups with a spatial grid resolution of $\Delta x = 300\,\text{m}\,(= 0.01\,d_e)$ and $N_x = 1000$ cells were initialized with an electron number density of $n_{\text{e}} = 0.4 \cdot 10^6\,\text{m}^{-3}$, an electron temperature of $T_{\text{e}} = 2.5\,\text{MK}$, and a magnetic field magnitude of 50 nT. In one simulation, the magnetic field direction was chosen to coincide with the extended simulation direction (resulting in parallel plasma wave modes to be resolved), in the other one, the magnetic field was set up perpendicular to the long dimension, resulting in perpendicular mode resolution. The plasma had zero bulk velocity in the simulation frame, with an added white noise velocity fluctuation of $\tilde{v} = 1000\,\text{m}/\text{s}$. The simulation was run for 0.037 seconds ($433\,\omega_{\text{pe}}^{-1}$).

Figure 4 shows the dispersion data resulting from spatial and temporal Fourier transform (using a von Hann window). Overlaid are analytic dispersion curves for the Langmuir wave (black dashed curve) and electron Bernstein modes (black solid curves). The wave behaviour in the simulation shows good agreement in both parallel and perpendicular directions. One noteworthy additional feature visible in the parallel direction (Figure 4a) is the presence of an entropy wave feature at low wave number $k$ and angular frequency $\omega$ that shows a quantization consistent with the electron velocity space resolution.

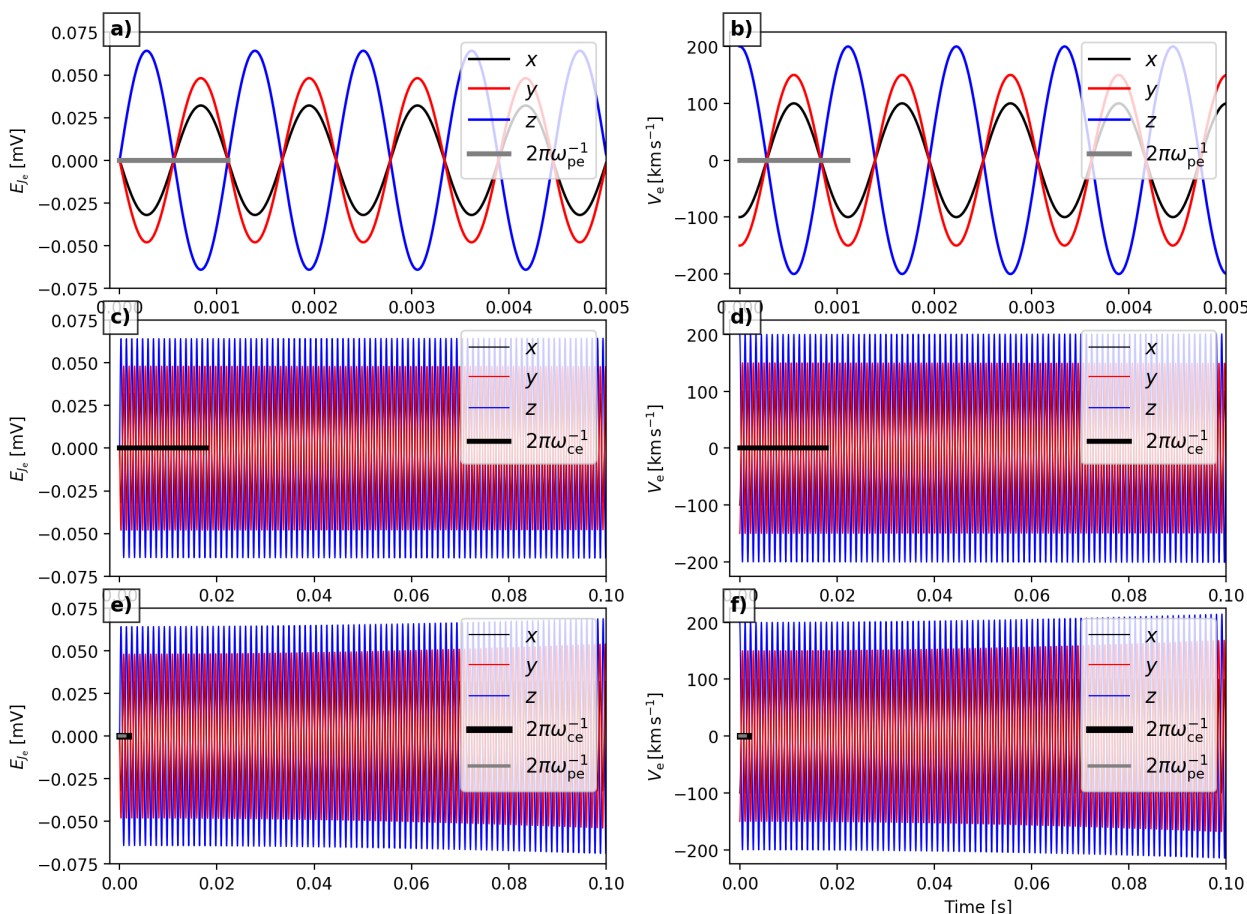

**Figure 3.** Graphs of solver stability in relation to electron plasma oscillation and gyromotion in a single-cell simulation. Note the different time axes used. Panels a), c), and e): Oscillation electric field $\boldsymbol{E}_{J_e}$ components. Panels b), d), and f): Electron bulk velocity $\boldsymbol{V}_e$ components. Panels a) and b) graph values in relation to the electron plasma oscillation period (indicated with a thick grey bar) and panels c) and d) in relation to the electron gyroperiod (indicated with a thick black bar), with a background magnetic field of $B = 20\,\mathrm{nT}$. Panels e) and f) showcase a simulation with a magnetic field of $B = 200\,\mathrm{nT}$, resulting in the gyro- and oscillatory motions interacting over multiple periods.

### 4.3 Stability within larger simulation domain

We also evaluate the stability of our solver over the larger simulated domain described in Section 3.4, with initialization values derived from the Vlasiator hybrid-Vlasov simulation. These graphs are shown in Figure 5. Panels a through e show the evolution of electron temperature values over a simulation of $1.0$ s, covering hundreds of electron plasma periods and, for the most part, tens of gyroperiods. We evaluate minimum, maximum, mean, and median values for total, $\boldsymbol{B}$-parallel, and $\boldsymbol{B}$-perpendicular electron temperatures. The system is seen to relax somewhat towards a final state, though some evolution is still apparent at

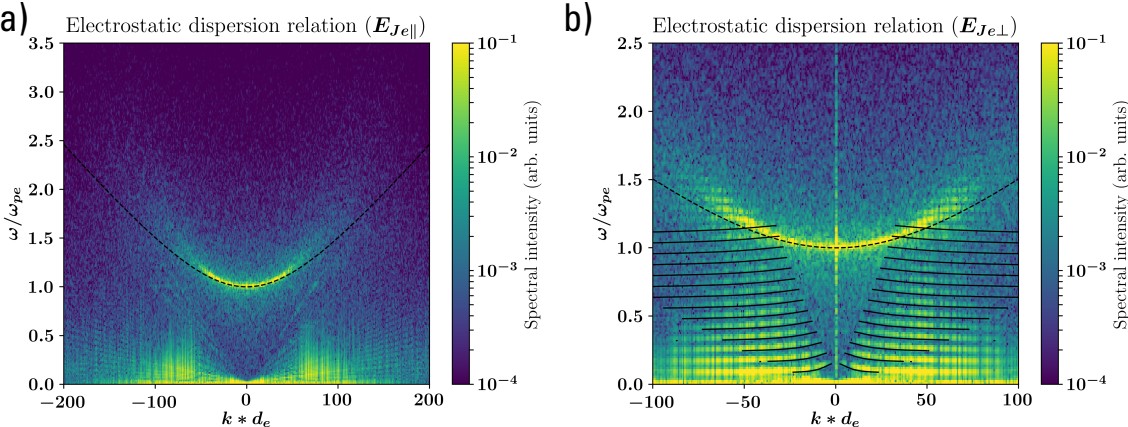

**Figure 4.** Dispersion analysis of the electron solver in a 1D test case with an axis-parallel (panel a) and axis-perpendicular (panel b) magnetic field. The colormap shows the spatiotemporal Fourier transform of $E_{J_e,\parallel}$ (panel a) and $E_{J_e,\perp}$ (panel b) overlaid with analytical solutions for the Langmuir wave (black dashed curve) and Bernstein modes (black solid curves).

the end of the simulation, possibly due to boundary effects. The maximum temperature plot in panel b is of particular interest as the hottest plasma cells appear to diffuse into their surroundings until $t \sim 0.4$ s when dynamic gyration processes overtake this temperature diffusion with perpendicular heating.

Panel f shows the agyrotropy measure (Swisdak, 2016) calculated from the electron pressure tensor, indicating that in the majority of the simulation domain electrons remain gyrotropic and even peak values do not grow past $10^{-3}$. Panel g shows statistics for the electron number density deviation from the initialisation value, indicating loss of plasma neutrality due to the motion of electrons. The minimum value oscillating between approximately $10^{-9}$ and $10^{-6}\,\mathrm{cm}^{-3}$ indicates the level of numerical fluctuations, and the maximum, mean and median values show how charge imbalance does grow initially but

stabilises within about $0.1\,\mathrm{s}$ and does not grow beyond $10^{-1}\,\mathrm{cm}^{-3}$.

In panels h through k of Figure 5 we show how the instantaneous plasma oscillation electric field $\boldsymbol{E}_{J_e}$ is well-behaved throughout the simulation box, converging towards stable values. We note that as the $\boldsymbol{E}_{J_e}$ field oscillates around zero, the averages are indeed zero throughout (not shown) and the values used for inferring minimum, maximum, mean and median values are instantaneous values from a arbitrary phase of the oscillation. In panel l we show the normalized current density $\boldsymbol{J}$

departure from the balance current $\boldsymbol{J_B} = \frac{\nabla \times \boldsymbol{B}}{\mu_0}$ which would be required to maintain the magnetic field structure according to Ampère's Law (4). This metric is seen to also stabilize, mostly at values well below unity. We expect the maximum value outliers to be due to locally small values of $\boldsymbol{J_B}$. Panels m and n show statistics for the parallel and perpendicular components of the electric field caused by electron pressure gradients, that is, the $-\frac{\nabla \cdot \mathcal{P}_e}{n_e e}$ term. As expected due to the ability of electrons to propagate along field lines, perpendicular components are much larger than parallel components. All components remain

stable at roughly their initial values. A minimum value is not shown as the use of a numerical slope limiter in the calculation of pressure gradients gives identically zero field components at local extrema of pressure.

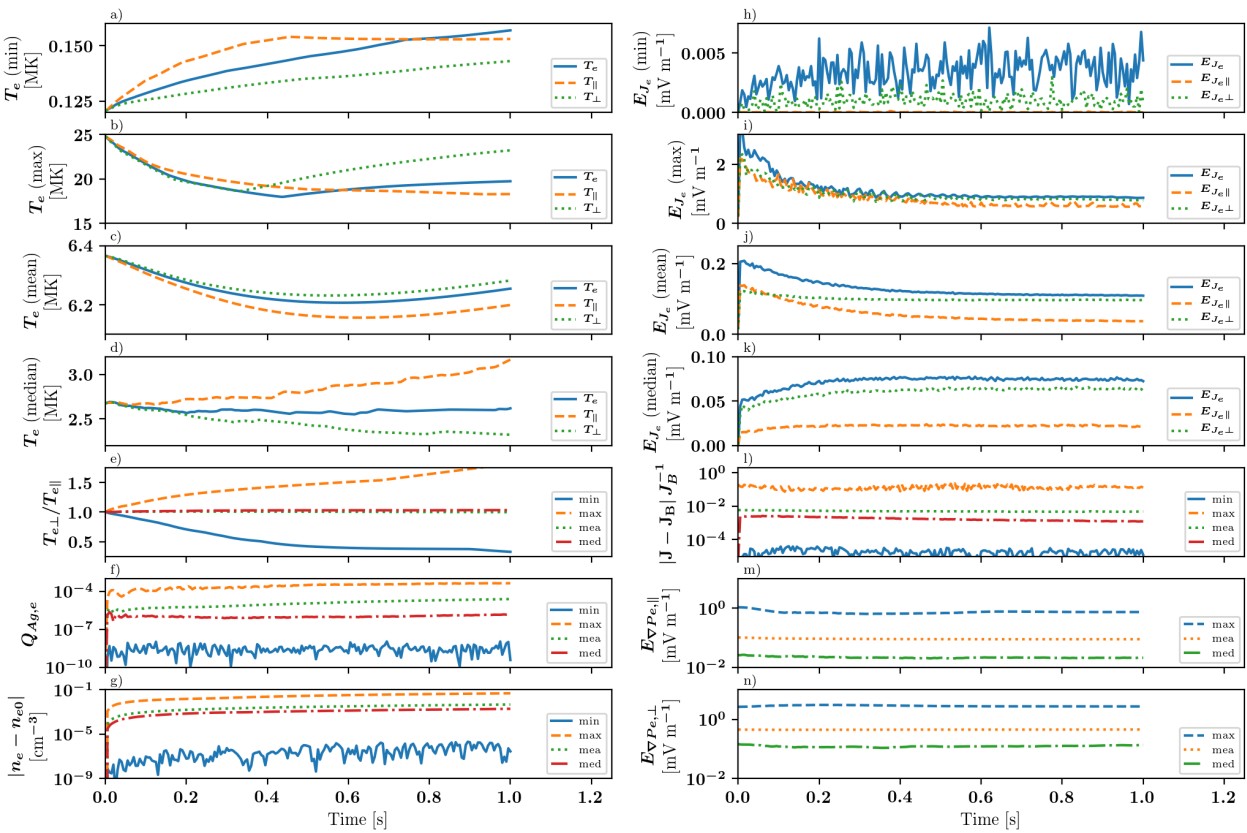

**Figure 5.** Evolution of electron and solver parameters over the whole simulation domain. **a–d**: Minimum, maximum, mean, and median values for electron temperature $T_e$ and its components parallel and perpendicular to the local magnetic field. **e**: Minimum, maximum, mean, and median values for electron temperature anisotropy. **f**: Minimum, maximum, mean, and median values for electron agyrotropy $Q_{\mathrm{Ag,e}}$. **g**: Minimum, maximum, mean, and median values for electron density deviation from initial state, indicating charge imbalance. **h–k**: Minimum, maximum, mean, and median values for the plasma oscillation electric field $\boldsymbol{E}_{J_e}$ and its components parallel and perpendicular to the local magnetic field. **l**: Minimum, maximum, mean, and median normalized values for current density $\boldsymbol{J}$ deviation from the value $\boldsymbol{J}_{\mathbf{B}} = \frac{\nabla \times \boldsymbol{B}}{\mu_0}$ required to fulfill Ampère's Law for the local magnetic field. **m,n**: Maximum, mean, and median values for parallel and perpendicular components of the electric field due to electron pressure gradients.

As part of our evaluation of solver stability, we performed a comparison run where our electron solver performed the rotation transformation corresponding with gyromotion in the Hall frame instead of in the substep-associated electron bulk frame. This transformation choice resulted in unstable growth of, in particular, $\boldsymbol{E}_{J_e}$, as could be expected (not shown).

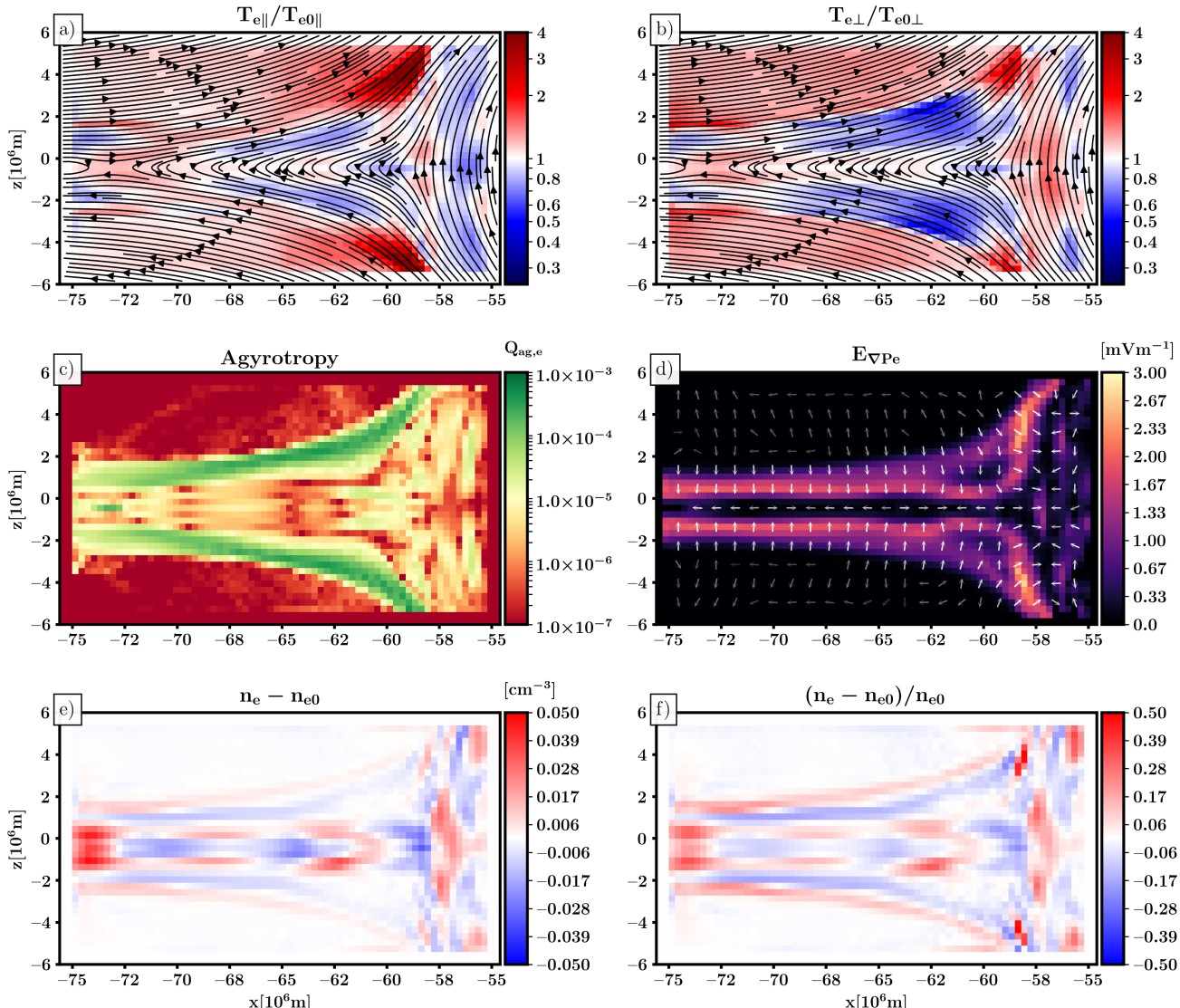

**Figure 6.** Electron distribution properties within the test domain after 1.0 s of simulation. **a**: The ratio of parallel electron temperature at 1.0 s to the parallel temperature at the start of the simulation, indicating parallel heating. **b**: The same but for perpendicular temperature. **c**: The agyrotropy measure for the electron population. **d**: The magnitude and direction of the electron pressure gradient term of the electric field. **e** and **f**: The charge imbalance $n_e - n_{e,0}$ and relative charge imbalance $(n_e - n_{e,0})n_{e,0}^{-1}$ found at the end of the simulation.

## 5  Results

Results from the electron simulation after $1.0$ s of evolution are presented in Figure 6. Figures 6a, b show parallel and perpendicular acceleration or deceleration of electrons as the ratio of end-of-simulation temperature to initial temperatures. Heating is

found in particular near the X-line configuration and where the plasma sheet boundary layer (PSBL) meets the magnetosphere, with parallel heating more localized than perpendicular heating.

Figure 6c shows the agyrotropy measure (Swisdak, 2016) calculated for electrons, indicating where the electron distribution has become non-gyrotropic. In most of the simulation domain, the value is very small, but enhanced agyrotropy (still relatively small values below $10^{-3}$) are found in the PSBL regions and at the magnetic field X-line. Some of this agyrotropy may be due to spatial sampling of electron gyromotion with a magnetic field gradient leading to larger gyroradii further away from the plasma sheet.

Figure 6d shows the electric field due to $\nabla \mathcal{P}_e$, with the field strongest where the PSBL meets the magnetosphere. The field direction is pointed towards the tail sheet or the magnetosphere, as expected. Magnitudes remain of the order of a few millivolts per metre.

Figures 6e,f quantify the charge imbalance resulting from electrons evolving due to static magnetic fields and the electric field resulting from the Ohm's Law terms presented in this paper. Figure 6e shows the level of charge imbalance as change in electron number density, and Figure 6f as the change scaled by the original electron number density. In the majority of the simulation domain, imbalance remains below $10^{-2}\,\mathrm{cm}^{-3}$. The electric field response is unable to maintain full plasma neutrality with some regions near the magnetosphere showing greater deviation from the initial state. Some stronger imbalance at the domain edges is likely a boundary effect which shall resolve itself with a larger simulation domain.

In Figure 7 we display electron velocity distribution functions after 1.0 s of simulation. Figure 7a shows the evolved electron temperature anisotropy $T_{\perp,e}T_{\parallel,e}^{-1}$, and Fig. 7b displays the maximum of instantaneous values of $\boldsymbol{E}_{J_e}$, taken over 10 measurements at 0.05 s intervals near the end of the simulation. Panels c) through n) of Figure 7 show parallel and perpendicular projections of electron eVDFs at virtual spacecraft (VSC) [1] through [6], with positions of VSC indicated in panels a) and b).

Figure 7a shows how temperature anisotropy $T_{\perp,e}T_{\parallel,e}^{-1}$ indicates parallel energization in the low-density regions adjacent to the PSBL and perpendicular energization adjacent to the X-line and within the tailmost region of the magnetosphere. As we have bulk flows of both ions and electrons towards the tail current sheet, some small part of this heating can be attributed to betatron acceleration as electrons convect towards stronger magnetic fields just adjacent to the actual high-beta plasma sheet. Other effects causing anisotropies may arise from spatial leakage of electrons undergoing plasma oscillation, with gyromotion binding perpendiculary heated electrons to the oscillation region and parallel accelerated electrons propagating along field lines to the near-magnetosphere PSBL regions.

The maximum of instantaneous values of $\boldsymbol{E}_{J_e}$, shown in Figure 7b, indicate that the strongest electron oscillations on our simulated scales are found in or near the PSBL, which would be consistent with observations of electron-driven waves in the PSBL (Onsager et al., 1993). Some increase in $\boldsymbol{E}_{J_e}$ is seen also at the X-line location, but not in other parts of the current sheet. We note that the X-line included in the Vlasiator simulation snapshot was not actively reconnecting. Comparison with Figure 7a and virtual spacecraft measurements indicate that parallel features, akin to electron beams, are indeed found in regions with enhanced $\boldsymbol{E}_{J_e}$.

The temperature anisotropies found in the near-Earth tail region of our simulation are mostly in the $0.5\ldots1.5$ range. Artemyev et al. (2014) reported on Cluster observations of electron temperature anisotropies ranging from $0.8\ldots1.6$ and cen-

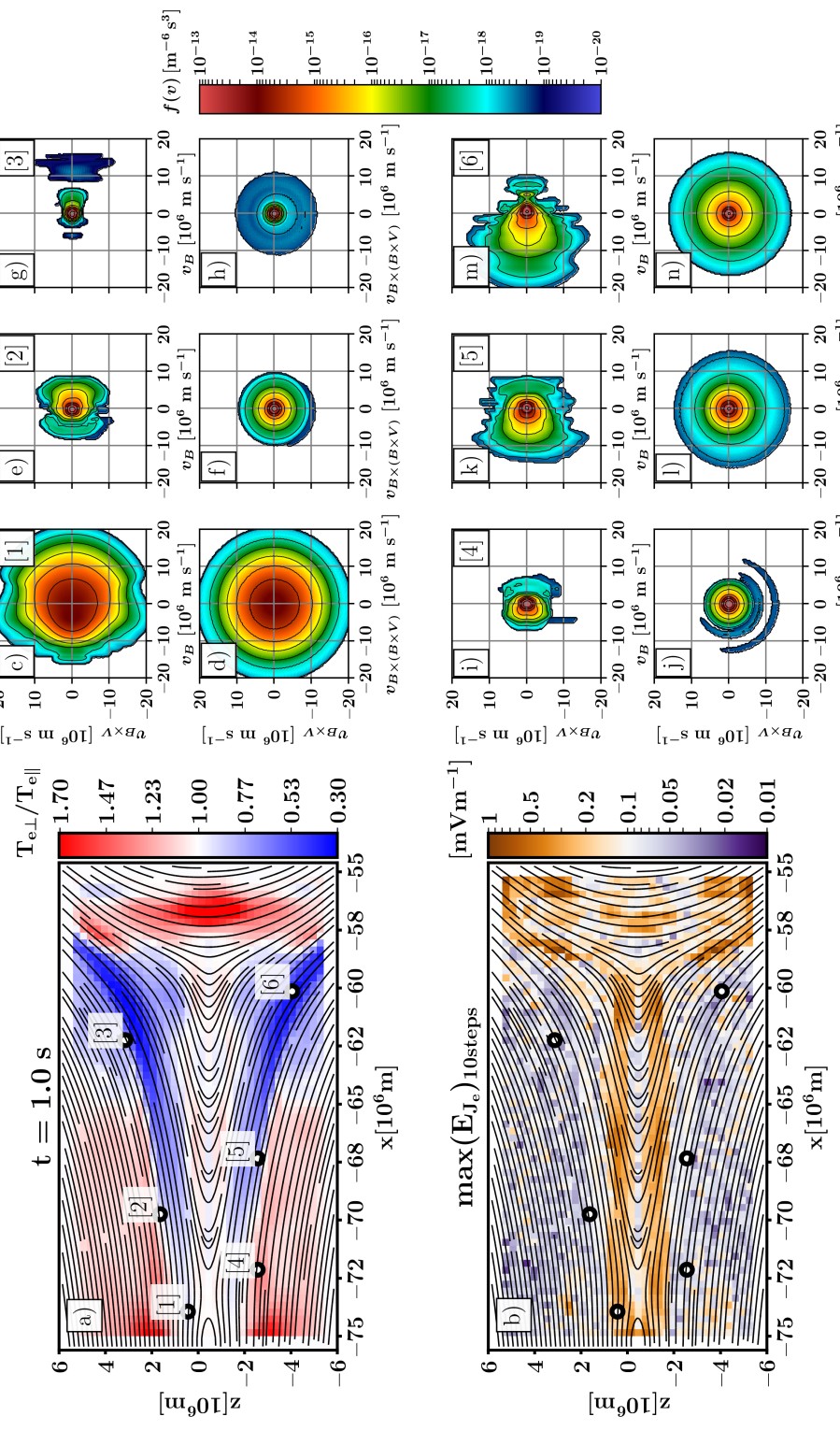

**Figure 7.** Electron properties and velocity distribution functions after 1.0 s of simulation. Panel a) Electron temperature anisotropy $T_{\perp,e}T_{\parallel,e}^{-1}$ overlaid with magnetic field lines and six virtual spacecraft locations, labelled [1]–[6]. Panel b): Maximum value for displacement current $E_{J_e}$, taken over 10 measurements at 0.05 s intervals near the end of the simulation. Panels c) through n): electron velocity distribution function projections into the parallel $v_B$ and perpendicular $v_{B\times(B\times V)}$ and $v_{V\times B}$ planes. Each virtual spacecraft is indicated by the number in the parallel eVDF panel with the panel below showing the corresponding perpendicular eVDF for the same virtual spacecraft.

tered around $\sim 1.1$, in agreement with our results, though their observations were gathered between $-20\,R_{\mathrm{E}} < x < -15\,R_{\mathrm{E}}$ $(-127 \cdot 10^6 < x < 96 \cdot 10^6\,\mathrm{m})$. Regions where parallel temperatures dominate (anisotropy $< 1$) are found in regions of cold
plasma, as can be seen by comparing Figures 2c and 7a. This does not preclude the possibility of parallel acceleration in regions of hot plasma, but rather shows that the acceleration may not be strong enough to be discerned over the main hot eVDF.

Parallel heating near the magnetotail plasma sheet has been reported to coincide with bi-directional electron distributions (Hada et al., 1981) with temperature ratios going up to 2–3, as in our simulation. Our VSC [2] and [5] show clear bi-directional
distributions. Due to our static background magnetic field, our parallel heating cannot be due to conventional Fermi accelera- tion. However, Hada et al. (1981) propose that adiabatic plasma processes where curvature drifts dominate over gradient drifts (Yamamoto and Tamao, 1978) can lead to significant parallel heating. Our VSC [1] is from close to the X-line and shows parallel elongation of the central part of the distribution, reminiscent of the football or shifted-football distributions of Figure 2 of Hoshino et al. (2001).

Asano et al. (2006) describe streaming 500 eV electrons at the PSBL, associated with a substorm event and variation of $B_y$, especially at small scales. Scaling with our electron mass, this corresponds to approximately $4000\,\mathrm{km\,s^{-1}}$ electron velocities, which is reasonably within the range of our eVDFs in Figure 7. We note that our simulation produces a background $B_y$ profile with $\nabla B_y$ in agreement with Figure 4 of Asano et al. (2006) (not shown), on top of which the streaming electrons are observed. Onsager et al. (1991) describe a simple 2-D Liouville model for the PSBL, as well as some ISEE-1 and ISEE-2
observations supporting their model. The formation mechanisms of eVDFs in Onsager et al. (1991) are listed as time-of-flight, energy conservation and magnetic moment conservation, which are included in our model, though we perform a more robust evaluation of plasma oscillation interplay with gyration. The eVDFs shown in their Figure 4 agree with e.g. our VSC [1], [2], [5], and [6]. We also note our VSC [3] displaying a disjoint parallel beam, matching the ISEE-2 observations in Figure 5 of Onsager et al. (1991).

Observations of perpendicular crescents are shown in MMS data in Burch et al. (e.g. 2016b, 2019) at EDRs, in conjunction with dayside magnetopause reconnection sites. These observed structures are produced at very small spatial scales, not cap- tured by our current model. We do, however, observe similar agyrotropic crescents in our results further out (in particular in Figure 7j), suggesting successful capture of a level of electron dynamics. These perpendicular crescents are found at very low phase-space density values, as could be expected by the low agyrotropy values seen in Figure 6e. Something akin to a parallel
electron crescent (Burch et al., 2016b) can be seen in Figure 7c, and bi-directional distributions as reported in Figures 6 and 7 of Burch and Phan (2016) are qualitatively similar to our Figures 7k and m.

## 6   Conclusions

In this method paper we have presented a novel approach to investigating electron distribution function dynamics in the context of global ion-hybrid field structures. Our method exploits global dynamics provided by hybrid-Vlasov simulations in order to
evaluate the response of gyrating and plasma oscillating electrons to global magnetic field structures.

We have shown our solver to behave in a stable manner, resolving electron inertia and updating a responsive electric field $E_{J_e}$ derived from the displacement current. If run at much finer spatial resolutions, our model replicates Langmuir waves and electron Bernstein modes. Electron temperatures evolve in response to the field structure but do not experience uncontrolled growth. Our sample simulation produces multiple features associated with spacecraft observations of eVDFs, such as parallel acceleration, bi-directional distributions, and perpendicular crescents.

Our model has several built-in limitations as it does not treat electrons as a fully self-consistent species. Magnetic fields gathered from the Vlasiator simulation are kept constant and thus force electron bulk motion to adhere to the required current density structure. As the initialization information is gathered from a hybrid-Vlasov simulation, it has a spatial resolution far below that required for resolving electron-scale waves such as whistlers, Bernstein waves and chorus waves. Scattering of electrons via these missing waves is somewhat accounted for by initializing every simulation from a Maxwellian isotropic distribution. These features together limit the applicability of the model to short periods of time. On the other hand, our model is efficient, and much larger spatial domains of investigation are easily achievable. Also, multiple eVlasiator runs can be performed from a single Vlasiator magnetosphere run to evaluate different driving conditions such as temperature ratios and anisotropies. The method builds on the efficiently parallelized Vlasiator codebase and will benefit from future numerical and computational improvements to Vlasiator solvers.

Our model can be applied to investigate electron dynamics on global spatial scales, with the current version applicable to 2D investigations, e.g., in the noon-midnight meridional plane. Electron velocity distribution functions generated by the model can be used to investigate, e.g., energetic electron precipitation into the Earth's auroral regions. The generated electron anisotropies can be used to infer regions where, for example, whistler waves can be expected to grow. The model can be run for several different initialization time steps to evaluate long-term evolution of precipitating electron distributions. This could be used to, for example, evaluate electron distribution changes as bulk flows and dipolarization fronts in the Earth's magnetotail propagate earthward. Li et al. (2020) observe electron Bernstein modes driven by perpendicular crescent distributions. As we have shown in Figures 4 and 7, with sufficient resolution we can reproduce electron Bernstein waves and agyrotropic electron distributions. Thus, we are in a position to investigate this connection further in eVlasiator.

Future improvements to our model will allow simulation initialization from non-uniform 3D-3V Vlasiator meshes, allowing investigation of spatially three-dimensional topologies including tail plasma sheet clock angle tilt. A possible path of future investigation would be to upsample the initialization fields and moments in order to achieve better resolution, but we emphasize that the model does not attempt to solve electrons in a fully self-consistent manner as magnetic fields are still kept constant. Increasing resolution by interpolating the input moments to a finer grid might not significantly improve plasma sheet density and temperature profiles. Increasing spatial resolution introduces numerous caveats including increased computational cost and possible charge imbalance resulting from spatially resolved electron oscillations, though our dispersion tests did not indicate such problems. If such imbalances arise from a future model, some method of solving Gauss' Law such as a Poisson solver should be implemented. A more detailed investigation into comparing electron eVDFs and dynamics with observations is expected in a future study.

*Code and data availability.* Vlasiator (http://www.physics.helsinki.fi/vlasiator/, Palmroth, 2020) is distributed under the GPL-2 open source license at https://github.com/fmihpc/vlasiator/ (Palmroth and the Vlasiator team, 2020). Vlasiator uses a data structure developed in-house (https://github.com/fmihpc/vlsv/, Sandroos, 2019). The Analysator software (https://github.com/fmihpc/analysator/, Battarbee and the Vlasiator team, 2020) was used to produce the presented figures. The run described here takes several gigabytes of disk space and is kept in storage maintained within the CSC – IT Center for Science. Data presented in this paper can be accessed by following the data policy on the
Vlasiator web site.

*Author contributions.* MB wrote the manuscript and code description. MB and TB devised the solver method. MA assisted with data analysis, model development, and comparisons with observations. UG performed the dispersion tests. MP is the PI of Vlasiator and leads the investigation. YPK, MG, KP, AJ, LT, MD, and MP participated in discussion and finalization of the manuscript.

*Competing interests.* The authors declare that they have no conflict of interest.

*Acknowledgements.* We acknowledge The European Research Council for Starting grant 200141-QuESpace, with which the Vlasiator model (http://www.physics.helsinki.fi/vlasiator) was developed, and Consolidator grant 682068-PRESTISSIMO awarded for further development of Vlasiator and its use in scientific investigations. We gratefully acknowledge Academy of Finland grants number 309937-TEMPO and 312351-FORESAIL. PRACE (http://www.prace-ri.eu) is acknowledged for granting us Tier-0 computing time in HLRS Stuttgart, where Vlasiator was run in the HazelHen machine with project number 2014112573 and in the Hawk machine with project number 2019204998.
The work of LT is supported by the Academy of Finland (grant number 322544). The authors wish to thank the anonymous referees for their assistance in improving the approachability of the manuscript.

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
