# Peer review of "Vlasov simulation of electrons in the context of hybrid global models: An eVlasiator approach"

_Annales Geophysicae, 2020_

## Referee Comment (RC1) · Anonymous Referee #1 · 18 Jun 2020

This is interesting draft describing inclusion of electron physics into the global hybrid simulations. Topic is very important, and such improved models are expected to provide useful and crucial information about many magnetopsheric plasma processes. Thus, paper should be published in AnGeo! However, some clarifications are needed before publication. I have one general suggestion, and set of specific comments.

Beside the reconnection, the electron physics in the magnetotail (the simulation domain shown in this study) includes: (1) electron adiabatic heating during earthward convection and transport (e.g., doi:10.1002/2015JA021166, 10.5194/angeo-31-1109-2013) (2) generation of electron anisotropy at plasma flow fronts and plasma injections and further relaxation of this anisotropy via whistler wave generation (e.g., doi:10.1103/PhysRevLett.106.165001, 10.1002/2016GL069188,

10.1029/2018GL079613) (3) formation of strong field-aligned and transverse electron currents in the magnetotail current sheet (e.g., doi:10.1029/2007JA012760, 10.1002/2016GL072011) (4) electron-ion decupling and formation of strong electric field in thin current sheets (e.g., doi:10.1029/2018JA026202, 10.1002/2016JA023325) (5) electron precipitation altering MI coupling (e.g., doi:10.1007/s11214-016-0234-7) It would be very useful to discuss which of these processes can be described by the presented model.

Comments: *Line 43: Do you mean "reconnection in Harris current sheet"? please, separate reference to the analytical model (Harris 1962) and numerical simulations

*Line 70: please, add reference to doi:10.1002/2015GL063946

*Lines 149-154: if I understand correctly, Authors exclude pressure gradient term, but include electron inertial term. This is quite unexpected solution. Ratio of electron inertial term and pressure gradient is of the order of Ve*R/T/Vt^2 where Ve is the bulk electron speed, R and T are typical spatial and temporal scales, and Vt is the electron thermal speed. To make this term much larger than one (neglect pressure versus inertia), one needs to consider processes with the evolution rate R/T»Vt^2/Ve i.e. much faster than electron thermal speed that is the largest speed in solar wind, magnetosheath, magnetotail plasmas. Authors should explain why they can use the Ve*R/T/Vt^2»1 assumption in the magnetosphere.

* Fig6a: Do Author suggest that this anisotropy results after 1s from the initially isotropic Maxwellian distribution? This time seems to be large in comparison with plasma timescales (inverse plasma frequency), but should be very small in comparison with global plasma/magnetic field motion responsible for betatron acceleration. Additional clarifications are needed here to explain how electrons can be heated transversely so quickly.

* Fig6a: I see Tper/Tpar ratio around 1.5, what is quite large ratio for magnetosphere. Do Authors observe whistler wave generation by anisotropic electrons and following relaxation of this anisotropy?

[Figure]

*Line 299: electron-scale waves at PSBL are driven by electron beams from the reconnection region. Do Authors observe such an acceleration?

*Lines 301-304: note, typical electron anisotropy in the magnetotail Tpar/Tper>1 is formed by cold (subthermal) electron populations. Is this the case in simulation?

*Fig6, velocity distributions: almost all shown distributions demonstrate a certain nongyrotropy (although weak): non-circle shape in v_perp1, v_perp2 plane. Such nongyrotropy is expected in the close vicinity to the reconnection region, but should be explained outside of this region where electrons are well magnetized .

---

## Short Comment (SC1) · 19 Jun 2020

I have read this paper with much interest, since it promises to solve a long-standing problem in plasma simulations, namely the large separation of scales between ions and electrons. However, I have found this paper disappointing and completely unclear in the descriptions of the numerical algorithm. For me, the main question remains: How can you follow the electron VDF in a hybrid model? Either you have kinetic electrons or you don't.

This approach does not seem to follow neither the Darwin approximation, nor the neutral vlasov approach proposed in https://doi.org/10.1063/1.4907665

So how do they do it??

---

## Referee Comment (RC2) · Anonymous Referee #2 · 9 Jul 2020

This draft described a Vlasov solver for electrons. This electron Vlasov solver is implemented to work with Vlasiator, which is a Vlasov-hybrid (kinetic ions and fluid electrons) code, together. This electron solver is distinct from a typical Vlasov solver in two ways: 1) the initial plasma and electromagnetic fields are initialized from the Vlasiator simulation results and the magnetic field is fixed during the electron simulation, and 2) the electric field that is produced by the electron oscillation is taken into account for accelerating electrons. The fixed magnetic field limits the applicability of the model to short-time simulations. Including the electron oscillation electric field is a novel feature.

Including electron dynamics into Vlasiator is definitely import, and I think the result of this research project should be eventually published somewhere. However, this manuscript needs significant improvement before it can be accepted. This is a paper

presents the numerical algorithm for the electron solver. The numerical algorithm itself is not complicated at all, but the draft is not well-organized and it is extremely difficult for readers to understand the algorithm.

Specific comments:

1. In the introduction part, some descriptions about previous works are not accurate or even wrong. 1) In line 27, the papers cited are not particle-in-cell codes. They are hybrid codes, just as indicated by their titles. In the space physics community, 'PIC' means both electrons and ions are represented by macro-particles. 2)line 39: Resolving Debye length is required by typical explicit PIC, but not implicit PIC. Please make it clear. 3)line 41: 'at the cost of loss of some electron physics.' The cost comes from a coarse grid and large time step instead of the implicit solver itself. 4) line 63: '... a local six-moment...'. These high-order moments fluid codes can be used for global simulations. They are not 'local'. The authors may also want to cite the paper Wang, Liang, et al. "Comparison of multi-fluid moment models with particle-in-cell simulations of collisionless magnetic reconnection." Physics of Plasmas. 5) line 64: '...they do not capture reconnection'. What does 'not capture reconnection' mean? I cannot believe any high-order moments paper would make such a note. High-order moments methods go beyond Hall-MHD, and they are at least as good as Hall-MHD, which is already capable of producing some import reconnection features, such as the fast reconnection rate and the Hall magnetic fields. 6) line 52:'with a proton-electron mass ratio of 25'. 25 is just a parameter for a specific simulation. It is not a feature of a model.

2. Section 2 and section 3 need to be re-organized. Section 2.1 describes Vlasiator, which is not new and it should be a separate section. It is better to combine section 2.2 and Section 3, since both describe the electron solver algorithm. In this new electron solver section, the authors should discuss the big picture of the electron solver with a few sentences first, for example, the authors should emphasize 1) this electron solver is also a Vlasov solver, just like the ion part, but the electric field is different, and 2) the

initial condition settings. Then, the general approach (not just for a specific simulation!) of initialing the electron solver and the details of the numerical steps should be carefully described.

3. Line 140: 'we assume charge neutrality to hold as div(E) = net_charge = 0, This simplifies our electric field calculations significantly as we do not need to implement a Poisson electrostatic solver.' This statement is not correct. 1) You can assume the net charge is zero in the Vlasov-hybrid simulation, but not in a simulation with the electron solver because it is not guaranteed. You cannot assume div(E) = 0, because they are NOT equal. With Hall term, E = -V_e x B, and I do not think it is guaranteed div(-V_e x B) = 0. Actually, if you calculate div(E) in your 2D magnetosphere Vlasiator simulation, you may find div(E) is not zero somewhere. 2) You do not need a Poisson solver to keep div(E) = net_charge != 0 if eq (4) is solved properly.

4. Line 150: 'As it is a feature of only the hybrid approach, it is not included in our electron solver.' I do not understand this statement, why it is a feature of 'only the hybrid approach'?

5. People usually use the uppercase \Delta instead of the low case \delta to describe numerical schemes. The authors should clearly define what is \delta V_e with proper superscripts and subscripts. For example: \Delta V_{e,i}^n = V_{e,i}^{n+1} - V_{e,i}^{n}.

6. Why the RK4 scheme is chosen? Is not a 2nd-order scheme accurate enough for this purpose? Give an explanation, please.

7. Why do you need sub-stepping? Why the sub-stepping time step is so small (line 176)?

8. What is a 'transformation matrix'? It has never been defined.

9. In figure 3, it seems both the velocity and electric field are growing slowly. What if running the simulation longer, for example, 10s?

10. Line 343: 'our model is efficient, taking only 80 thousand CPU hours to perform

the sample simulation presented in this paper'. Without comparison, I cannot see why '80k CPU hours' is 'efficient'.

11. LIne 357: What is 'Upscaling the input moments'?

---

## Author Comment (AC1) · 26 Aug 2020

Response to review by Anonymous Referee #1

We wish to thank the referee for their input and evaluation of our manuscript. Below, we have included the referee comments in italics and our own response in regular text.

*This is interesting draft describing inclusion of electron physics into the global hybrid simulations. Topic is very important, and such improved models are expected to provide useful and crucial information about many magnetospheric plasma processes. Thus, paper should be published in AnGeo!*

Thank you, we agree it is an interesting topic where a lot of progress can be made!

[Figure]

*However, some clarifications are needed before publication. I have one general suggestion, and set of specific comments. Beside the reconnection, the electron physics in the magnetotail (the simulation do-main shown in this study) includes: (1) electron adiabatic heating during earth-ward convection and transport (e.g., doi:10.1002/2015JA021166, 10.5194/angeo-31-1109-2013 ) (2) generation of electron anisotropy at plasma flow fronts and plasma injections and further relaxation of this anisotropy via whistler wave generation (e.g., doi:10.1103/PhysRevLett.106.165001, 10.1002/2016GL069188, 10.1029/2018GL079613 ) (3) formation of strong field-aligned and transverse electron currents in the magnetotail current sheet (e.g., doi:10.1029/2007JA012760, 10.1002/2016GL072011 ) (4) electron-ion decoupling and formation of strong electric field in thin current sheets (e.g., doi:10.1029/2018JA026202, 10.1002/2016JA023325 )(5) electron precipitation altering MI coupling (e.g., doi:10.1007/s11214-016-0234_7 ) It would be very useful to discuss which of these processes can be described by the presented model.*

Thank you for the comprehensive suggestions and references to aid us in performing this evaluation. We shall expand the discussion regarding added references.

*Comments:*
*Line 43: Do you mean "reconnection in Harris current sheet"? please, separate reference to the analytical model (Harris 1962) and numerical simulations*

Yes, a good point, we shall separate them and clarify this section.

*Line 70: please, add reference to doi:10.1002/2015GL063946*

Thank you for the excellent suggestion.

*Lines 149-154: if I understand correctly, Authors exclude pressure gradient term, but include electron inertial term. This is quite unexpected solution. Ratio of electron inertial term and pressure gradient is of the order of $\frac{V_e R}{T V_t^2}$ where $V_e$ is the bulk electron speed, $R$ and $T$ are typical spatial and temporal scales, and $V_t$ is the electron thermal*

*speed. To make this term much larger than one (neglect pressure versus inertia), one needs to consider processes with the evolution rate $\frac{R}{T} \gg \frac{V_t^2}{V_e}$ i.e. much faster than electron thermal speed that is the largest speed in solar wind, magnetosheath, magnetotail plasmas. Authors should explain why they can use the $\frac{V_e R}{T V t^2} \gg 1$ assumption in the magnetosphere.*

Thank you for bringing this up. We agree that the ratio $\frac{V_e R}{T V_t^2}$ is not expected to be much larger than one within the domain under investigation. After further evaluation, we agree that assessing the electron pressure gradient term will likely be a good choice, and are in the process of adding the necessary modules to the code. Subsequently, the manuscript will be updated with this description.

*Fig6a: Do Author suggest that this anisotropy results after 1s from the initially isotropic Maxwellian distribution? This time seems to be large in comparison with plasma time-scales (inverse plasma frequency), but should be very small in comparison with global plasma/magnetic field motion responsible for betatron acceleration. Additional clarifications are needed here to explain how electrons can be heated transversely so quickly.*

As our initial distributions are indeed Maxwellian and isotropic, this does appear to be the case. We agree that betatron acceleration should not result in such changes at these time scales, but the interplay of drifts with electron oscillation appear to be behind this effect. We shall also add evaluation of how much of the seen effect is actual perpendicular or parallel acceleration, and how much is due to different temperatures of electrons convecting along field lines.

*Fig6a: I see $T_\perp/T_\parallel$ ratio around 1.5, what is quite large ratio for magnetosphere. Do Authors observe whistler wave generation by anisotropic electrons and following relaxation of this anisotropy?*

Thank you for the good question. Evaluation of different kind of waves (power and frequencies) generated by electrons within the target simulation domain is something that we would like to investigate in the future. However, our current implementation

approach is to maintain static magnetic fields, so the solver only captures electrostatic oscillations, which do not include whistler waves. Future expansion of the simulation code is planned to implement a more complete field solver, which could also capture whistler waves, which can then be reported in future publications.

*Line 299: electron-scale waves at PSBL are driven by electron beams from the reconnection region. Do Authors observe such an acceleration?*

Panel b of Figure 6 indicates in an orange color the regions where electron-scale oscillations visible via large values of $E_{J_e}$ are strongest, and panel a indicates regions where parallel electron pressure dominates as blue regions. Comparing these regions with the visible tail magnetic field structure indicates that electron oscillations are found throughout the PSBL region, even when pressure anisotropy is close to 1 (e.g. virtual spacecraft location 1). We do note however that at virtual spacecraft location 1 there does appear to be some parallel structure to electrons. We shall add additional figures of simulation results to visualise and exemplify the resultant dynamics.

*Lines 301-304: note, typical electron anisotropy in the magnetotail $T_\parallel/T_\perp > 1$ is formed by cold (subthermal) electron populations. Is this the case in simulation?*

Yes, comparison of electron temperature and temperature anisotropy plots (Figures 2c (proton temperatures but scales with electron temperatures and 6b, respectively) confirms that $T_\parallel/T_\perp > 1$ is associated with cold electron populations.

*Fig6, velocity distributions: almost all shown distributions demonstrate a certain non-gyrotropy (although weak): non-circle shape in $v_{\perp 1}$, $v_{\perp 2}$ plane. Such non-gyrotropy is expected in the close vicinity to the reconnection region, but should be explained outside of this region where electrons are well magnetized.*

The registered values of agyrotropy (Swisdak 2016) are indeed nonzero but still low, remaining below $5.0 \times 10^{-4}$ everywhere in the simulation domain. Please see the attached Figure 1. We see these moderate values of gyrotropy both at the magnetic

reconnection topology site, and in the PSBL. The effect within the PSBL is most likely due to some hot electrons originating within the tail plasma sheet performing gyro-motion which, after they have entered the lower **B** region in the PSBL, causes them to spread in the perpendicular direction. We shall investigate this further and see if finetuning our solver parameters changes this effect.

[Figure]

[Figure]

**Fig. 1.**

---

## Author Comment (AC2) · 26 Aug 2020

We wish to thank the referee for their input and evaluation of our manuscript. Below, we have included the referee comments in italics and our own response in regular text.

*This draft described a Vlasov solver for electrons. This electron Vlasov solver is implemented to work with Vlasiator, which is a Vlasov-hybrid (kinetic ions and fluid electrons) code, together. This electron solver is distinct from a typical Vlasov solver in two ways: 1) the initial plasma and electromagnetic fields are initialized from the Vlasiator simulation results and the magnetic field is fixed during the electron simulation, and 2) the electric field that is produced by the electron oscillation is taken into account for accelerating electrons. The fixed magnetic field limits the applicability of the model to*

*short-time simulations. Including the electron oscillation electric field is a novel feature. Including electron dynamics into Vlasiator is definitely import, and I think the result of this research project should be eventually published somewhere. However, this manuscript needs significant improvement before it can be accepted. This is a paper presents the numerical algorithm for the electron solver. The numerical algorithm itself is not complicated at all, but the draft is not well-organized and it is extremely difficult for readers to understand the algorithm.*

Thank you for the constructive criticism. We will strive to improve the presentation of the work, as indeed explanation and understanding of the method is what we wish to achieve.

*Specific comments:*
*1. In the introduction part, some descriptions about previous works are not accurate or even wrong. 1) In line 27, the papers cited are not particle-in-cell codes. They are hybrid codes, just as indicated by their titles. In the space physics community, 'PIC' means both electrons and ions are represented by macro-particles.*

We believe there may be different sub-understandings of these terms, as we are familiar with terms hybrid-PIC and full-PIC to differentiate between these two approaches. Both approach still include particles tracked across cells. We will clarify the terms in this manner.

*2) line 39: Resolving Debye length is required by typical explicit PIC, but not implicit PIC. Please make it clear.*

This was noted in the sentence starting on line 40, but we agree it can be misread and shall rewrite this to be more clear.

*3) line 41: 'at the cost of loss of some electron physics.' The cost comes from a coarse grid and large time step instead of the implicit solver itself.*

A good point, this shall be clarified.

*4) line 63: '... a local six-moment...'. These high-order moments fluid codes can be used for global simulations. They are not 'local'. The authors may also want to cite the paper Wang, Liang, et al. "Comparison of multi-fluid moment models with particle-in-cell simulations of collisionless magnetic reconnection." Physics of Plasmas.*

Thank you for the excellent suggestion and the correction. The referenced six-moment code was only presented via local cases, but we shall include references to global multiple-moment codes as well.

*5) line 64: '...they do not capture reconnection'. What does 'not capture reconnection' mean? I cannot believe any high-order moments paper would make such a note. High-order moments methods go beyond Hall-MHD, and they are at least as good as Hall-MHD, which is already capable of producing some import reconnection features, such as the fast reconnection rate and the Hall magnetic fields.*

This was written in response to the conclusions of the referenced Huang 2019 paper. We shall correct this section to correctly describe a wider range of multiple-moment codes.

*6) line 52: 'with a proton-electron mass ratio of 25'. 25 is just a parameter for a specific simulation. It is not a feature of a model.*

That is correct, but it was the parameter used in the simulation used in that publication. We shall rewrite this sentence to clarify this fact.

*2. Section 2 and section 3 need to be re-organized. Section 2.1 describes Vlasiator, which is not new and it should be a separate section. It is better to combine section 2.2 and Section 3, since both describe the electron solver algorithm. In this new electron solver section, the authors should discuss the big picture of the electron solver with a few sentences first, for example, the authors should emphasize 1) this electron solver is also a Vlasov solver, just like the ion part, but the electric field is different, and 2) the initial condition settings. Then, the general approach (not just for a specific simulation!)*

*of initialing the electron solver and the details of the numerical steps should be carefully
described.*

Thank you for these suggestions. After consideration, we agree that moving section 2.2
into section 3 and adding initial descriptive text is a good choice. We will also provide
more details regarding the initialisation and boundary conditions, as well as implement
naming convention clarifications.

*3. Line 140: 'we assume charge neutrality to hold as div(E) = net charge = 0, This
simplifies our electric field calculations significantly as we do not need to implement a
Poisson electrostatic solver.' This statement is not correct. 1) You can assume the net
charge is zero in the Vlasov-hybrid simulation, but not in a simulation with the electron
solver because it is not guaranteed. You cannot assume div(E) = 0, because they
are NOT equal. With Hall term, $E = -V_e \times B$, and I do not think it is guaranteed
div($-V_e \times B$) = 0. Actually, if you calculate div(E) in your 2D magnetosphere Vlasiator
simulation, you may find div(E) is not zero somewhere. 2) You do not need a Poisson
solver to keep div(E) = net charge $\neq$ 0 if eq (4) is solved properly.*

Thank you for these comments. Just to clarify, we did not intend to imply we actually
constrained div(E) to zero, (or a given value of $\rho_q$) but rather that we chose to assume
that charge imbalances generated during this short simulation period would remain
small, and thus, the electric field contribution due to them could be neglected. To
probe this issue, we are in the process of investigating charge imbalance resulting
from running our electron code. We shall discussion to this effect and quantify the
magnitude of charge imbalance forming due to electron effects.

We acknowledge that a suitably well performing full-Maxwellian field solver should also
be able to correctly model effects due to charge imbalance, and intend to investigate
this in a future update of our model.

*4. Line 150: 'As it is a feature of only the hybrid approach, it is not included in our
electron solver.' I do not understand this statement, why it is a feature of 'only the*

*hybrid approach'?*

This approach stemmed from the quasi-neutrality assumption, but upon further reflection, we have decided to implement an electron pressure gradient term into the solver after all. We shall add description of this term and results into the manuscript.

*5. People usually use the uppercase $\Delta$ instead of the low case $\delta$ to describe numerical schemes. The authors should clearly define what is $\delta V_e$ with proper superscripts and subscripts. For example: $\Delta V_{e,i}^n = V_{e,i}^{n+1} - V_{e,i}^n$.*

In our approach we designated uppercase $\Delta$ as effects happening on the full grid level with lowercase $\delta$ steps being performed in substepping on a cell-by-cell basis. We shall add description and clarification in order to rectify these issues.

*6. Why the RK4 scheme is chosen? Is not a 2nd-order scheme accurate enough for this purpose? Give an explanation, please.*

Correct, the stability of the substepping is quite demanding. We initially investigated using Runge-Kutta-Nyström schemes, but upon testing found that the relatively simple and flexible RK4 scheme provided best results. The computational price of RK4 within this context is minimal in comparison with the Vlasov advection computations.

*7. Why do you need sub-stepping? Why the sub-stepping time step is so small (line176)?*

Each remapping of the gridded electron (or proton) distribution function (be it rotation, acceleration or advection) involves piecewise fitting of polynomials to small sections of the distribution function and integrating over sections of them. This is computationally expensive and also, if performed needlessly often, can lead to numerical diffusion. Also, after each full simulation time step (consisting of advection and acceleration remapping steps), we need to perform communication with other processes. These together indicate that any calculations which can be substepped, should be. We shall add discussion about this approach to the manuscript to clarify the issue.

*8. What is a 'transformation matrix'? It has never been defined.*

We apologise for this oversight. It is used to evaluate acceleration of the gridded distribution function used in the Slice-3D solver Vlasiator approach (combining rotation and field-parallel acceleration into one descriptive matrix which is then decomposed into three shear motions). We shall add description to this effect.

*9. In figure 3, it seems both the velocity and electric field are growing slowly. What if running the simulation longer, for example, 10s?*

We performed additional tests, running the single-cell tests for longer periods of time (1s, matching our target scenario). Indeed, oscillations begin to increase, but we were able to negate this by decreasing the RK4 substep length, maintaining stability even over extended periods of time. However, the growth is significant only when $\Omega_{ce}\Omega_{pe}^{-1} \approx 1$, which does not occur within our simulation domain. In the future, when we apply this method to larger domains, this validity needs to be ensured, or the substep length needs to decreased accordingly. We shall add discussion of this stability issue to the manuscript.

*10. Line 343: 'our model is efficient, taking only 80 thousand CPU hours to perform the sample simulation presented in this paper'. Without comparison, I cannot see why '80k CPU hours' is 'efficient'.*

We acknowledge that this point is perhaps not the most informative, but indeed, comparisons of similar electron approaches are not readily available. We shall amend the statement.

*11. LIne 357: What is 'Upscaling the input moments'?*

We were referring to potentially performing interpolation of proton input moments in order to increase the resolution of simulation initialisation values. We shall clarify this discussion.

---

## Author Comment (AC3) · 26 Aug 2020

*I have read this paper with much interest, since it promises to solve a long-standing problem in plasma simulations, namely the large separation of scales between ions and electrons. However, I have found this paper disappointing and completely unclear in the descriptions of the numerical algorithm. For me, the main question remains: How can you follow the electron VDF in a hybrid model? Either you have kinetic electrons or you don't. This approach does not seem to follow neither the Darwin approximation, nor the neutral vlasov approach proposed in https://doi.org/10.1063/1.4907665 So how do they do it??*

Thank you for your interest in our work. We would like to temper expectations in that we

do not propose to solve the scale separation issue, but instead offer a new simulation method for evaluating certain aspects of electron dynamics within a plasma environment generated by a hybrid model. We also acknowledge that certain facets of this code shall be improved upon in the future, but that is the nature of all simulation codes.

We found the neutral Vlasov approach an interesting read, and note that it indeed takes a different approach, investigating the low-frequency limit. We commend the convergence approach taken in that paper. Our paper does not aim to supersede that method, but rather provide a complementary approach, focusing the investigation on high-frequency electron oscillations within the simulated magnetic domain.

We shall strive that the clarified revision of our manuscript shall be clearer in how our model relates to existing codes and approaches. However, we respectfully refrain from designating electron kinetics into a binary categorization - it is possible to model different aspects of electron kinetics, always making some compromises along the way.

---

## Author Response (AR2)

**Vlasov simulation of electrons in the context of hybrid global models: An eVlasiator approach**

**Response to anonymous referee #2, 1.12.2020**

**This manuscript has been improved significantly. However, it is still difficult to understand the algorithm that is described in section 3.2.**

*Thank you for the detailed assistance in improving the understandability of the manuscript. We have modified the structure and added more description of the details of the method. We decided to not repeat the actual implementation of the SLICE-3D transformation, as that was not changed from regular Vlasiator, but have added a more direct reference to Palmroth et al 2018 (Vlasov methods in space physics and astrophysics) with a reference to the section detailing the procedure. Please note that the html version of that manuscript does not contain section numberings, but they are available in the PDF version.*

**The first half of section 3.2 is good (line 190-215), but the second half is confusing. It seems the authors start describing the sub-stepping algorithm before describing the big picture of the whole algorithm. Actually, figure 1 is very informative. If the authors can describe the solver from up to down following figure 1, it will be much easier for readers to understand. Here, I provide my personal suggestions to organize the second half of section 3.2.**
**1) Describe what is known at the start of a cycle time=t (6D electron phase space distribution at t? EM field at t?) and what is calculated from time=t to time=t+\Delta t.**
**2) Describe the leapfrog algorithm ('translation' and 'acceleration'). I strongly suggest the authors writing down some expressions to explain the scheme. For example, even though the authors have added a sentence to explain what is a 'transformation matrix' in line 155, it is still not straightforward for me to understand the role of the matrix.**
**3) Explain why a substepping approach is required and why the substepping time step is so small. Simply saying 'in order to accurately model the electron gyromotion and plasma oscillation' does not explain too much.**
**4) Explain why a 4th order scheme is necessary and describe how the substepping scheme works.**

**The authors do not have to follow the steps strictly but I would like to see something similar. It seems step 4 is described but steps 1-3 are missing.**

*As requested, we have rewritten the description along the suggested lines. We admit that describing the transformation matrix generation and application may still be challenging to grasp, but hope that our references to our general Vlasov methods paper and the sub-stepping details will prove helpful. We acknowledge that the numerical requirements for substepping are very strict with numerous RK4-substeps, but we did perform convergence tests to arrive at this requirement, and have mentioned this in the manuscript.*

**Minor comments:**

**1) Line 41: '...though the resolution decrease incurs the loss of some electron physics'. This is not fair to the implicit methods. The implicit PIC can use a coarse grid but they do not have to. When an implicit PIC code uses a fine grid resolution as an explicit PIC, they contain the same physics.**

*We have further edited this section to clarify that it is the resolution decrease, not the implicit solver, which results in loss of some electron physics.*

**2) Line 217: the number '22' is like a magic number, why it is not '21' or '23'? I guess there is a range for the time step. Please provide more explanation.**

*We have elaborated that this value has been chosen based on stability analysis of VDF evolution, balanced against computational cost. Each semi-Lagrangian remapping is quite expensive so we want to do as large steps as possible, but going much above 22 degrees per rotation is not resolved well by the SLICE-3D algorithm. We note that semi-Lagrangian remapping methods are far superior to Eulerian algorithms for this rotation in terms of time step length.*

**3) Section 3.1: how are the ions handled? Do they keep the initial values and do not change during the simulation?**

*Yes, we have elucidated that ions are kept static. Their motion could be evaluated, but should be minimal within the short 1 second simulation length, so we have chosen to only propagate electrons which are the species of interest.*

[revised manuscript text omitted]